# Time-resolved NMR monitoring of tRNA maturation

Pierre Barraud [1], Alexandre Gato[1], Matthias Heiss[2], Marjorie Catala[1], Stefanie Kellner [2] & Carine Tisné [1]

Although the biological importance of post-transcriptional RNA modifications in gene expression is widely appreciated, methods to directly detect their introduction during RNA biosynthesis are rare and do not easily provide information on the temporal nature of events. Here, we introduce the application of NMR spectroscopy to observe the maturation of tRNAs in cell extracts. By following the maturation of yeast tRNA[Phe] with time-resolved NMR measurements, we show that modifications are introduced in a defined sequential order, and that the chronology is controlled by cross-talk between modification events. In particular, we show that a strong hierarchy controls the introduction of the T54, Ψ55 and m$^1$A58 modifications in the T-arm, and we demonstrate that the modification circuits identified in yeast extract with NMR also impact the tRNA modification process in living cells. The NMR-based methodology presented here could be adapted to investigate different aspects of tRNA maturation and RNA modifications in general.

[1] Expression génétique microbienne, UMR 8261, CNRS, Université de Paris, Institut de biologie physico-chimique, 13 rue Pierre et Marie Curie, 75005 Paris, France. [2] Department of Chemistry, Ludwig Maximilians University Munich, Butenandtstr. 5-13, 81377 Munich, Germany. Correspondence and requests for materials should be addressed to P.B. (email: pierre.barraud@cnrs.fr) or to C.T. (email: carine.tisne@cnrs.fr)

Over 170 nucleotide modifications are currently reported in RNAs from the three domains of life, the vast majority being found in tRNAs[1,2]. This family not only displays the largest variety of post-transcriptional decorations among RNA molecules, but also the highest density of modifications per RNA transcript[3,4]. The introduction of post-transcriptional chemical modifications is central in the maturation process to generate functional tRNA molecules[5–7]. The biogenesis of tRNAs is a multi-step process that comprises the synthesis of a tRNA precursor (pre-tRNA) by transcription of a tRNA gene, the removal of the 5'-leader and 3'-trailer sequences, the addition of the 3'-CCA amino-acid accepting sequence, the removal of intronic sequences when present, and the incorporation of a large number of post-transcriptional chemical modifications[8]. Modifications are found in two main regions of the L-shaped structure of tRNAs, the tRNA core, and the anticodon-loop region (ACL)[4]. All cellular functions of tRNAs, including their alternative functions outside translation, are controlled and modulated by modifications[5,9,10]. While modified nucleotides in the ACL participate in decoding during protein synthesis[11,12], those in the tRNA core are collectively implicated in the folding and stability of tRNAs[13].

While a simple model would be that modifications are introduced to tRNA independently, several modification circuits have been identified in which one or more modifications stimulate formation of a subsequent modification[7,14]. This obviously drives a defined sequential order in the tRNA modification process. In addition, the interdependence of multiple modification events might in certain cases be responsible for the variations of tRNA modifications in response to environmental perturbations[2,7,15–17]. The characterization of modification circuits is therefore central to understand the dynamic regulation of modifications in tRNAs, however their identification remains difficult, since monitoring tRNA maturation at a single nucleotide level in a time-resolved fashion is technically challenging[18].

In the past 10 years, NMR spectroscopy became a pre-eminent method to investigate post-translational modifications (PTMs) in proteins[19,20]. Time-resolved NMR measurements provided the means to monitor the establishment of PTMs in vitro, in cellular extracts and in living cells[21–25]. In addition, monitoring the introduction of PTMs with NMR has provided mechanistic insights into modification hierarchies, with initial modifications exhibiting stimulatory or inhibitory effects on subsequent modification events[26,27].

Here, inspired by the NMR monitoring of PTMs in cellular environments, we report an original methodology to monitor RNA modifications in cellular extracts with NMR. Using yeast tRNA$^{Phe}$ as a model system, we demonstrate that multiple modification events can be monitored in yeast extract with NMR in a time-resolved fashion. Using continuous NMR measurements to measure a series of snapshots of the tRNA along the maturation route, we observe a sequential order in the introduction of several modifications. This suggests that modification circuits could control the tRNA$^{Phe}$ maturation process in yeast. We next adopted a reverse genetic approach and investigated the interplay between the different modifications in yeast tRNA$^{Phe}$ with both NMR and mass spectrometry and show that modification circuits identified in the yeast extract on tRNA$^{Phe}$ also influence the process of tRNA modification in living cells.

## Results

**Monitoring RNA modifications in cellular extracts with NMR.** In order to investigate the different steps along the maturation pathway of post-transcriptionally modified RNAs, we sought to implement NMR-based methods for monitoring the introduction of modifications in RNA substrates. As a general concept, we believed that modification reactions by RNA-modifying enzymes could be probed by NMR in cell extracts containing enzymatic activities responsible for the modification of the RNA substrate of interest. Introducing isotope-labeled RNAs into unlabeled cell extracts combined with the use of isotope-filters in NMR experiments enables the detection of the sole RNA of interest within the complex cell extract environment. The non-disruptive nature of NMR provides the means to directly monitor RNA modification events in a continuous and time-resolved fashion, by measuring successive NMR experiments on a single sample (Fig. 1a). In addition, since NMR spectroscopy provides information at atomic-resolution, multiple RNA modifications introduced on the same substrate, e.g. methylations on nearby nucleotides, can be easily distinguished.

Our approach for monitoring of RNA post-transcriptional modifications with NMR relies on the fact that imino signals of RNAs are very sensitive to their chemical environment. Imino groups are carried by uridines and guanosines, and are easily observed in $^{1}$H–$^{15}$N correlation spectra on condition that the imino proton is protected from exchange with the solvent by hydrogen bonding in, e.g. a Watson–Crick-like pairing. Different types of behavior for imino signals can in principle be observed in $^{1}$H–$^{15}$N correlation spectra upon enzymatic modifications (Fig. 1b). The incorporation of a chemical group on a defined nucleotide will affect the chemical environment of the modified nucleotide itself but also of nearby nucleotides. Overall, this will cause the progressive disappearance of signals from the unmodified RNA and the correlated appearance of new signals from the modified RNA. For the sake of clarity, the disappearance and the correlated appearance related to the imino signal of a nucleotide experiencing a chemical modification will be thereafter called 'a direct effect' (signal M on Fig. 1b), whereas the disappearance and the correlated appearance associated with the imino signal of a nearby nucleotide will be referred to as 'an indirect effect' (signal P on Fig. 1b). The possibility to observe indirect effects enables the detection of modifications on adenosines and cytosines, even though they do not carry imino groups.

**Yeast tRNA$^{Phe}$ as a model for monitoring RNA modifications.** As a proof of concept, we undertook the investigation of the maturation of the yeast tRNA$^{Phe}$ in yeast extract. Matured tRNA$^{Phe}$ contains 14 modified nucleotides (Fig. 2a and Supplementary Table 1), among which: (i) T54, Ψ55, and D16 are found in almost all yeast tRNAs; (ii) m$^{1}$A58, m$^{2}_{2}$G26, m$^{2}$G10, Ψ39, m$^{7}$G46, and m$^{5}$C49 are frequently found in yeast tRNAs; and (iii) Gm34, yW37, and m$^{5}$C40 are found uniquely in tRNA$^{Phe}$ in yeast[1]. To evaluate the ability of yeast extracts to modify yeast tRNA$^{Phe}$, we first produced an unmodified $^{15}$N-labeled tRNA$^{Phe}$ by in vitro transcription (Fig. 2a), which to a certain extent can be regarded as a tRNA$^{Phe}$ precursor (pre-tRNA$^{Phe}$) with processed 5′- and 3′-termini and spliced intron, but lacking all RNA modifications. This choice of substrate affects the introduction of certain modifications near the anticodon[28], but constitutes a simplified system, which has the advantage to spotlight nucleotide modifications over RNA processing steps. The chemical shifts of imino groups involved in secondary and tertiary interactions were assigned using standard NMR procedures for RNAs[29]. NMR-fingerprints of tRNAs can be efficiently acquired with $^{1}$H–$^{15}$N Band-selective excitation short-transient Transverse relaxation optimized spectroscopy (BEST-TROSY) experiments[30]. To serve as a reference spectrum of yeast tRNA$^{Phe}$ without any modifications, we measured a $^{1}$H–$^{15}$N BEST-TROSY experiment in vitro (Fig. 2b), in a buffer aiming to approach cellular conditions. We next incubated this tRNA at 30 °C for 12 h in yeast cellular extracts prepared under mild conditions in order to preserve most enzymatic activities of the cell. Cellular extracts

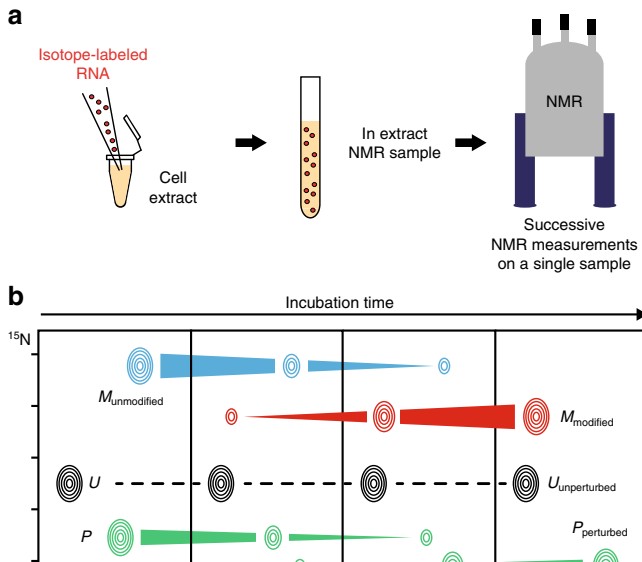

**Fig. 1** Schematic description of the method. **a** An isotope-labeled RNA (represented as red dots) is introduced in cell extracts in an NMR tube to yield an in extract NMR sample. Successive NMR measurements on a single sample incubated directly in the NMR spectrometer can then be performed. **b** RNA modifications are monitored on the RNA imino signals with 2D ($^1$H,$^{15}$N) NMR spectra in a time-resolved fashion. Modifications cause the progressive disappearance of imino signals from unmodified nucleotides (signal M disappearing in blue) and the correlated appearance of new imino signals from modified nucleotide (signal M appearing in red). Modifications also cause disappearance and correlated appearance of imino signals of nearby nucleotides (signal P in green). Nucleotides far away from modification sites have unperturbed imino signals over time (signal U in black)

were supplemented with enzymatic cofactors, such as S-adenosyl-L-methionine (SAM), the almost universal methyl donor of RNA methylations by methyltransferases[31] (MTases), and reduced nicotinamide adenine dinucleotide phosphate (NADPH), a hydride donor implicated in the reduction of uridines to dihydrouridines[32]. After the 12 h of incubation, we measured a BEST-TROSY experiment (experimental time of 2 h) directly in yeast extracts. The $^{15}$N-isotope filter of the experiment enables the exclusive detection of the $^{15}$N-labeled tRNA$^{Phe}$ signals, even in this complex environment (Fig. 2b). The comparison of this spectrum with the reference spectrum revealed obvious differences, with apparent additional signals, as well as duplicated or shifted signals (Fig. 2b). Remarkably, tRNA signals are still observed after this long incubation, confirming that tRNA transcripts are not rapidly degraded by RNases down to single nucleotides and are stable for several hours in yeast extracts[28]. In addition, the line-width of the NMR signals is not particularly broadened in the extracts, meaning that the overall tumbling time of the $^{15}$N-labeled tRNA$^{Phe}$ is unchanged compared with the in vitro situation, and therefore that it remains mostly free and does not predominantly associate with proteins or other cellular components within large molecular complexes. These two hurdles, namely substrate degradation and signal line broadening, were identified in a pioneer study as the most prominent difficulties that complicate in depth investigations of nucleic acids with NMR in cellular environments[33]. We did not face these difficulties in the case of tRNA transcripts in yeast extracts, which opened the way to a thorough investigation of the fate of tRNAs throughout their maturation pathway with NMR.

Changes in the NMR-fingerprint of tRNA$^{Phe}$ (Fig. 2b) can be associated with expected chemical modifications. For instance the signal of U55, a residue that gets modified into Ψ55 and that is found near two other modified nucleotides, i.e. T54 and m$^1$A58, gave rise to three individual signals (Fig. 2a, b). Another example comes from the signal of G24, which is split into two (Fig. 2b), and for which the appearance of the second signal could be linked to the m$^2$G10 modification, since the G24:C11 base pair stacks over this modified nucleotide (Fig. 2a). However, the exact assignments of all the changes on the NMR-fingerprint of tRNA$^{Phe}$, which are potentially generated by direct and indirect effects from all modifications, remains impractical only from the signal assignments of the initial unmodified tRNA transcript. We thus performed further investigations by NMR in order to accurately interpret these changes on the NMR-fingerprint of tRNA$^{Phe}$ upon incubation in yeast extracts.

**The NMR signature of individual modifications**. To characterize the effect of modifications on the NMR-fingerprint of tRNA$^{Phe}$, we measured NMR spectra on three distinct tRNA$^{Phe}$ samples differing in their modification content. The first sample is the aforementioned tRNA$^{Phe}$ sample produced by in vitro transcription, which presents none of the modifications (Fig. 3a, left). The second sample corresponds to native tRNA$^{Phe}$ purified from *Saccharomyces cerevisiae*, and therefore contains all 14 modifications found in fully modified yeast tRNA$^{Phe}$ (Fig. 3a, right). The third sample corresponds to a recombinant yeast tRNA$^{Phe}$ produced in and purified from an *Escherichia coli* strain overproducing this tRNA with a system previously described[34,35]. Since *E. coli* has a less prolific but related tRNA modification machinery than *S. cerevisiae*, the modification pattern of this sample resembles that of the fully modified yeast tRNA$^{Phe}$, but overall exhibits fewer modifications (Fig. 3a, center). The modification status of a tRNA expressed in heterologous systems can be inferred from the transposition of the known tRNA modification patterns found in the host organism onto the recombinant tRNA sequence. This approach predicts that yeast tRNA$^{Phe}$ produced in *E. coli* contains eight modifications, six of which are identical to the ones found in natural tRNA$^{Phe}$ (Fig. 3a). This modification pattern was confirmed by further NMR analysis of the sample.

We next performed the chemical shift assignments of the imino groups of the two modified tRNA$^{Phe}$ samples, produced in *E. coli* and yeast, with the same approach used for the unmodified tRNA$^{Phe}$ sample. Imino protons assignments of the fully modified tRNA$^{Phe}$ are consistent with the ones reported by early NMR studies of this tRNA[36,37]. Chemical shifts assignments of the imino groups of the three samples are reported on $^1$H–$^{15}$N BEST-TROSY experiments measured in identical conditions in Fig. 3b. The comparison of the NMR fingerprints of the three samples revealed the NMR signature of individual modifications (Fig. 3c). The identified direct effects and indirect effects are reported in Fig. 3c with solid line arrows and dashed line arrows, respectively. NMR signatures of modifications in the anticodon loop could not be identified on this type of NMR spectra, since their imino protons are not engaged in base pairing and are not protected from exchange with the solvent, and are therefore not detectable. For the identification of the NMR signature of modifications, the 3D structure of yeast tRNA$^{Phe}$ provided a valuable aid, enabling the precise identification of the nearest modifications to an imino group[38]. The identified NMR signatures (Fig. 3c) might seem puzzling if one looks at the 2D cloverleaf representation of tRNAs, but become more apparent if one has in mind the complicated 3D L-shaped structure of tRNAs (Supplementary Fig. 1). As an example, we describe a few NMR signatures of individual modifications derived from the

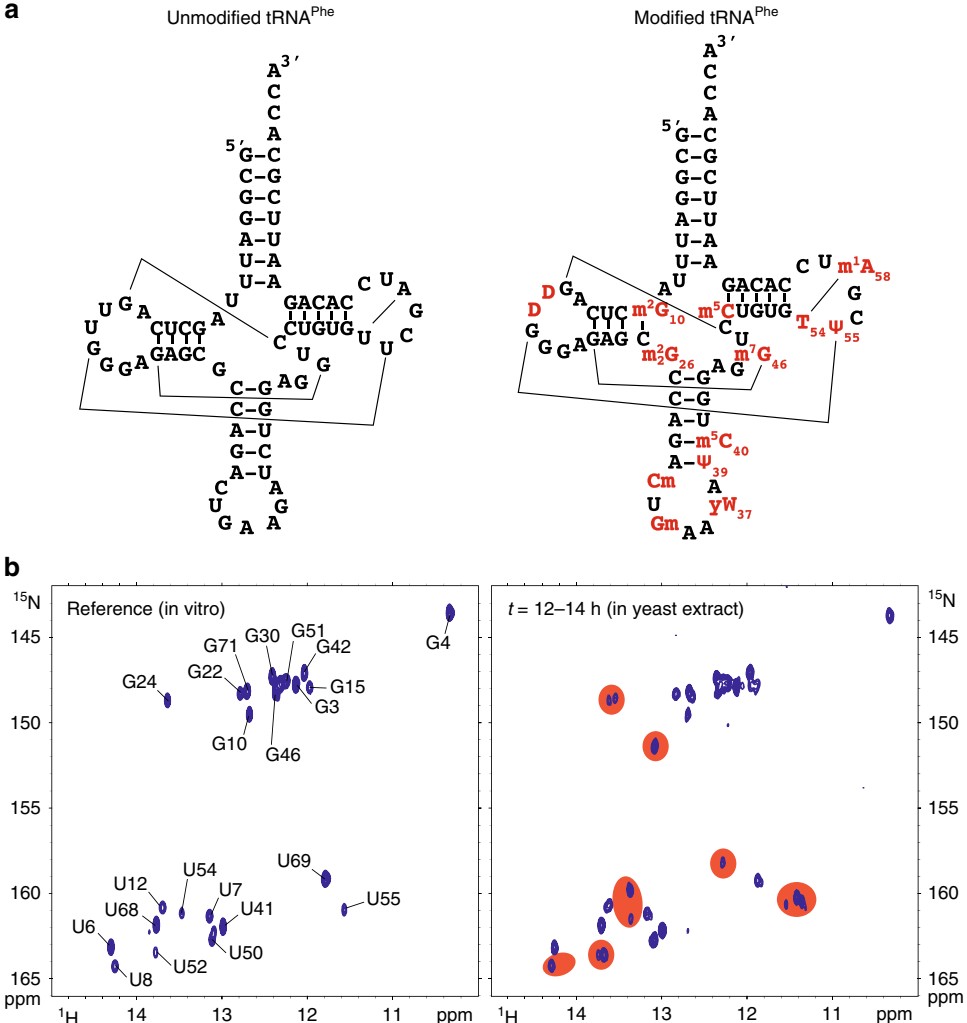

**Fig. 2** Modification of yeast tRNA^Phe in yeast extract. **a** Sequence and cloverleaf representation of unmodified yeast tRNA^Phe (left) and modified tRNA^Phe (right). Main tertiary interactions are represented with thin lines. Modifications are in red. **b** (left) Imino ($^1$H,$^{15}$N) correlation spectrum of a $^{15}$N-labeled unmodified tRNA^Phe measured in vitro in a buffer approaching physiological conditions (see "Methods" section). Assignments of the imino groups are reported on the spectrum. (right) Imino ($^1$H,$^{15}$N) correlation spectrum of a $^{15}$N-labeled tRNA^Phe measured after 12 h of incubation at 30 °C in yeast extract. The most obvious changes as compared with the spectrum of the unmodified tRNA^Phe are highlighted with red patches

comparison of the NMR fingerprints. For instance, the comparison of the unmodified with the *E. coli*-produced tRNA^Phe (Fig. 3c, left) reveals the NMR signature of the T54 modification, with a direct effect on the imino group of U54 (solid line) and an indirect effect on U52 (dashed line). Moreover, the comparison of the NMR-fingerprints of the three samples (Fig. 3c, right) reveals an NMR signature for the m$^1$A58 modification, with an indirect effect on the imino group of T54 (dashed line). The indirect effect is very pronounced in this case, most probably because T54 forms a non-canonical base pair with A58, the chemical environment of which is highly perturbed by the positive charged appearing on the A58 base after methylation at position 1. Similarly, the comparison of the unmodified with the *E. coli*-produced tRNA^Phe (Fig. 3c, left) reveals the NMR signature of the Ψ55 modification, with a direct effect on the imino group of U55 (solid line); and the comparison of the NMR-fingerprints of the three samples (Fig. 3c, right) reveals an NMR signature for the m$^1$A58 modification, with an indirect effect on the imino group of Ψ55 (dashed line). Overall, we identified the NMR signatures for nine modifications, most of which are located in the tRNA core, i.e. m$^2$G10, D16, and m$^2_2$G26 in the D-arm; m$^5$C40 in the anticodon-arm; m$^7$G46 in the variable region; and m$^5$C49, T54, Ψ55, and m$^1$A58 in the

T-arm (Fig. 3). This paved the way to a detailed investigation of the introduction of RNA modifications in yeast tRNA^Phe by NMR.

**Time-resolved NMR monitoring of RNA modifications**. In order to record time-resolved snapshots along the tRNA^Phe modification pathway, we incubated the unmodified $^{15}$N-labeled tRNA^Phe at a concentration of 40 μM in active yeast extract supplemented with the modification enzymes cofactors SAM and NADPH. The incubation was done at 30 °C directly in an NMR tube in the NMR spectrometer, and a series of $^1$H–$^{15}$N BEST-TROSY experiments were measured from initial mixing time up to ~24 h after starting the monitoring of tRNA maturation (Fig. 4a). At this concentration of $^{15}$N-labeled tRNA, reasonable signal-to-noise ratio (SNR) is achieved with an acquisition time of 2 h, and therefore each NMR fingerprint measurement spreads over a 2 h time period (Fig. 4a). Importantly, all changes observed in the NMR-fingerprints can be rationalized and attributed to specific RNA modifications on the basis of the NMR signatures identified above (Fig. 3). The identifications of specific RNA modification in each snapshot are displayed with solid line and

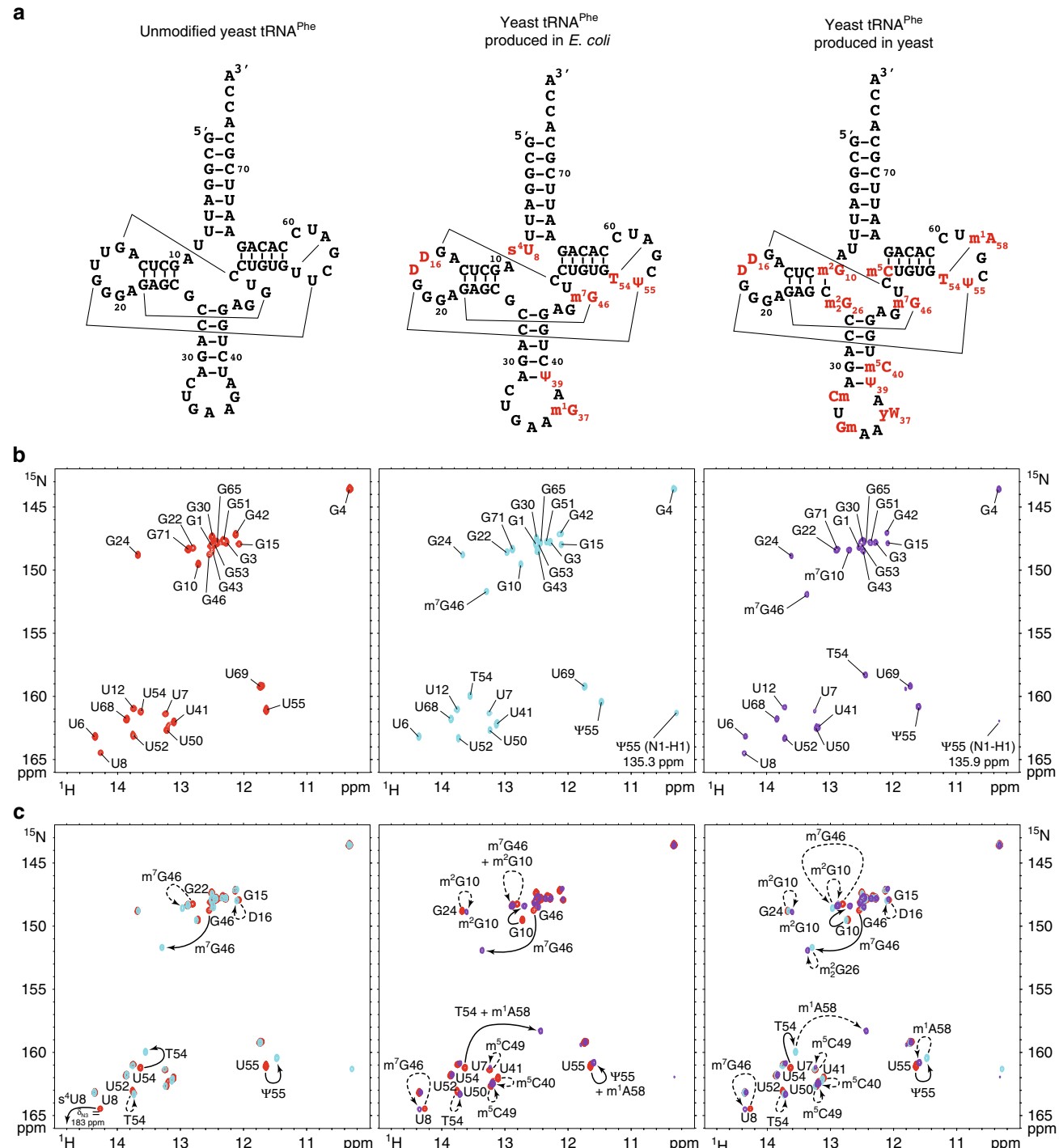

**Fig. 3** The comparison of different maturation states gives the NMR signature of individual modifications. **a** Sequence and cloverleaf representation of unmodified yeast tRNA^Phe produced by in vitro transcription (left), yeast tRNA^Phe produced and modified in *E. coli* (middle), and yeast tRNA^Phe produced and modified in yeast (right). Main tertiary interactions are represented with thin lines. Modifications are in red. **b** Imino ($^{1}$H,$^{15}$N) correlation spectra and imino group assignments of $^{15}$N-labeled tRNA^Phe measured in vitro: (left) unmodified yeast tRNA^Phe, spectrum in red; (middle) yeast tRNA^Phe produced and modified in *E. coli*, spectrum in cyan; (right) yeast tRNA^Phe produced and modified in yeast, spectrum in purple. **c** Superposition of imino ($^{1}$H,$^{15}$N) correlation spectra of various tRNA^Phe samples and NMR signature of individual modifications: (left) unmodified (red) and *E. coli*-modified (cyan); (middle) unmodified (red) and yeast-modified (purple); (right) unmodified (red), *E. coli*-modified (cyan), and yeast-modified (purple). A selection of imino assignments is reported on the spectra. NMR signatures of modifications are reported with continuous line arrows (direct effects—see main text), or dashed arrows (indirect effects)

dashed line arrows for direct and indirect effects, respectively (Fig. 4a). We observed different types of behavior for different modifications. First, some modifications appear early in the sequence, such as Ψ55, m$^{7}$G46, and T54; and some appear late, such as m$^{1}$A58. Second, some modifications are introduced during a short interval of time, such as Ψ55 and m$^{7}$G46, whereas others are introduced over a long time period, such as T54 and m$^{2}$G10, suggesting different levels of intrinsic catalytic activities

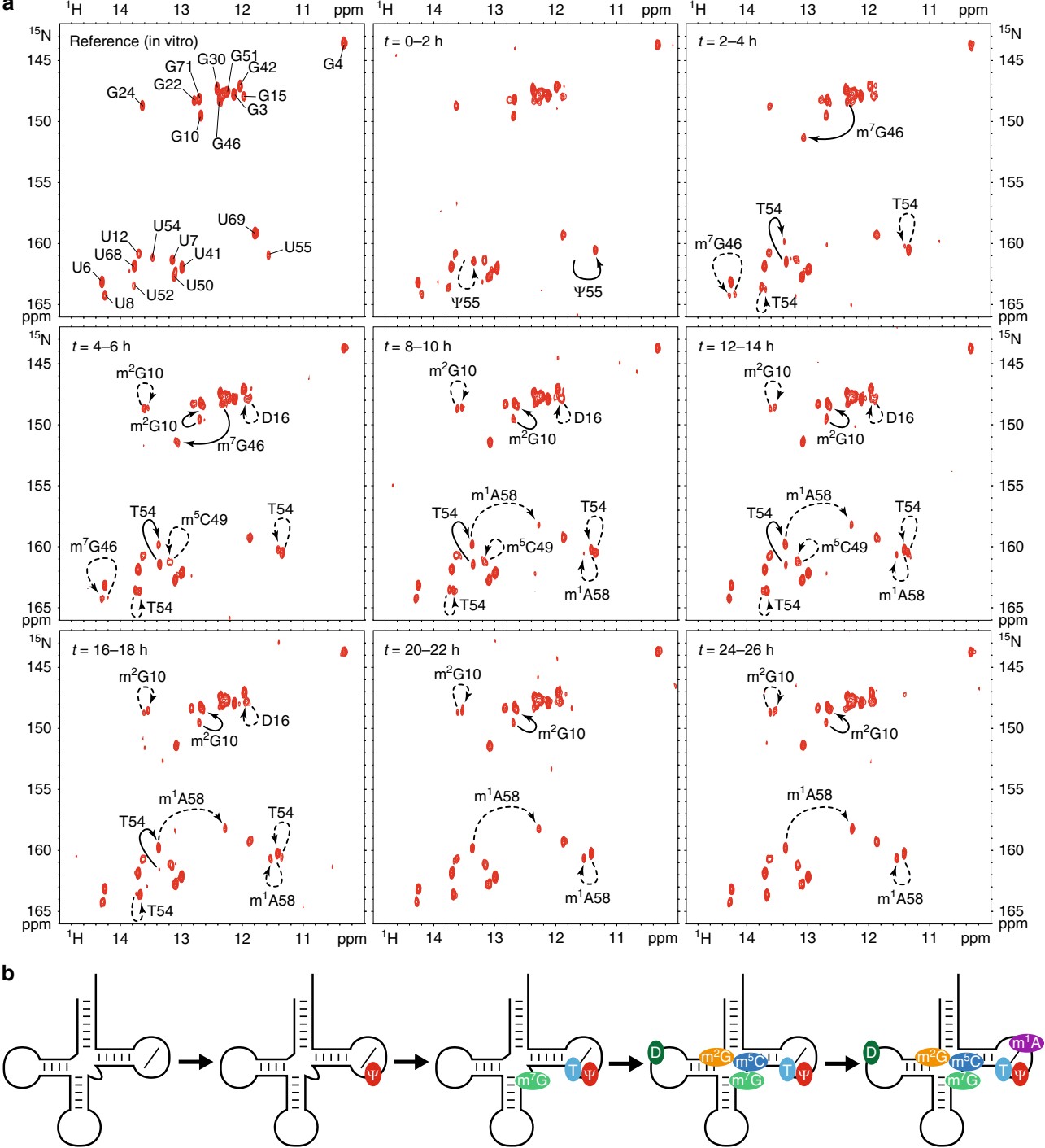

**Fig. 4** Time-resolved NMR monitoring of RNA modifications in yeast tRNA$^{Phe}$. **a** Imino ($^1$H,$^{15}$N) correlation spectra of a $^{15}$N-labeled tRNA$^{Phe}$ measured in vitro (first top left spectrum—same as Fig. 2b left) and in a time-resolved fashion during a continuous incubation at 30 °C in yeast extract over 26 h (remaining eight spectra). Each NMR spectrum measurement spreads over a 2 h time period, as indicated. Modifications detected at the different steps are reported with continuous line arrows for direct effects, or dashed arrows for indirect effects (see main text for the distinction between direct and indirect effects). **b** Schematic view of the sequential order of the introduction of modifications in yeast tRNA$^{Phe}$ as observed with NMR. Each modification is associated with a defined color also used in Figs. 5 and 6

for the different modification enzymes. In detail, the time-resolved NMR monitoring of RNA modifications in yeast tRNA$^{Phe}$ reveals that the modification Ψ55 is the first one to be introduced and that U55 is completely modified before any other modification can be detected (Fig. 4a, $t = 0$–2 h), showing that Pus4 is one of the most active modification enzymes in yeast and suggesting that it is highly efficient on a non-modified tRNA

transcript and does not require prior modifications. Then, m$^7$G46 and T54 are the two next modifications to be introduced (Fig. 4a, $t = 2$–4 h), and whereas m$^7$G46 is rapidly fully modified ($t = 2$–6 h), T54 is introduced over a longer time period ($t = 2$–18 h), suggesting that Trm2 is a less active enzyme than the Trm8–Trm82 complex. The three next modifications to be observed in the sequence are m$^2$G10, m$^5$C49, and D16 (Fig. 4a,

$t$ = 4–6 h). Among these three enzymes, the complex responsible for the introduction of the m$^2$G10 modification (Trm11–Trm112) appears to be the least active one. Even though the T54 modification is observed earlier in the time-course of modification, the three modified nucleotides m$^2$G10, m$^5$C49, and D16 are initially detected at a time point, where U54 is not completely converted into T54, meaning that these modifications could, in principle, be introduced on tRNAs lacking the T54 modification. The strict sequential introduction of m$^2$G10, m$^5$C49, and D16, only after T54 is fully present is also compatible with our data, since ensemble-averaged measurements do not allow us to discriminate between these two possibilities (see below). The last modification to be detected along the tRNA maturation pathway is the m$^1$A58 modification (Fig. 4a, $t$ = 8–10 h). Similarly, the m$^1$A58 modification is initially detected at a time point, where U54 is not completely modified into T54, meaning that it could, in principle, be introduced on tRNAs lacking the T54 modification. However, our data suggest that the m$^1$A58 modification is strictly introduced after T54. Indeed, we detect the m$^1$A58 modification by its indirect effect on the T54 imino group signal (Fig. 3c). If the m$^1$A58 modification is introduced on a tRNA lacking the T54 modification, it would be detected by a large perturbation on the U54 imino group that would then resonate at a different position than that of T54 ($\delta_H$ = 12.27 ppm; $\delta_N$ = 158.2 ppm), since it lacks the T54 modification. Importantly, no unidentified transient signal, which could correspond to tRNAs with m$^1$A58 but still lacking T54, is observed during the incubations. The m$^1$A58 modification of U54-containing tRNAs is therefore not observed in our experiments, suggesting a strict introduction of m$^1$A58 after T54.

As a summary, the sequence in the introduction of modifications in yeast tRNA$^{Phe}$ as monitored by NMR is schematically represented in Fig. 4b. Two modifications for which NMR signatures have been identified (Fig. 3c) were not observed with these conditions and incubation time, namely m$^2_2$G26 and m$^5$C40. The lack of the m$^5$C40 modification was expected, since it was reported to strictly depend on the presence of the intron at the level of the anticodon region[28]. Taken as a whole, we clearly observed a sequential order in the introduction of the modifications found in the heart of the 3D structure of yeast tRNA$^{Phe}$ (Fig. 4b), which could suggest that regulatory circuits are controlling the introduction of these modifications. However, since the modification content observed by NMR is averaged over the total tRNA$^{Phe}$ population, the exact modification status of each tRNA molecule cannot be firmly established. This type of questions that needs to be addressed at a single molecule level, cannot be resolved with currently available methods aimed at quantifying the modification content of RNAs. For instance, in our case it is not possible to know whether or not all m$^2$G10-containing tRNAs also contain T54 and m$^7$G46 modifications. Importantly, with our experimental setup, it is difficult to discriminate between an authentic dependence on the prior introduction of certain modifications and a sequential order caused by different intrinsic catalytic activities of the enzymes. In order to address these important questions concerning tRNA maturation and identify potential interdependencies among RNA modifications, we have adopted a reverse genetic approach and conducted a systematic analysis of tRNA$^{Phe}$ maturation in different yeast strains.

**Complex circuits of RNA modifications in yeast tRNA$^{Phe}$.** In order to identify a potential interplay between different modification enzymes, we recorded several time-resolved snapshots of the tRNA$^{Phe}$ modification pathway in identical conditions as in Fig. 4, with the exception that the unmodified $^{15}$N-labeled tRNA$^{Phe}$ was incubated in various extracts prepared from yeast strains depleted of one specific modification enzyme at a time (Fig. 5a and Supplementary Figs. 2–8). The modification pattern was then compared with that of the wild-type yeast extract (Fig. 4). First, this provided independent confirmation of the NMR signatures of the individual modifications identified previously, since changes in the NMR spectra associated with a given modification were no longer observed in the corresponding depleted yeast extract. Second, the detailed analysis of the different profiles of the tRNA$^{Phe}$ modification pathways, provided a way to visualize any subtle interdependence between the multiple modification events. To identify these correlated changes, it is best to compare the whole series of snapshots measured in the wild-type and depleted yeast extracts side by side (Fig. 4a and Supplementary Figs. 2–8). However, for ease of visualization, a single time point, intermediate in the time-course of the tRNA$^{Phe}$ modification pathway, and at which most of the differences between the wild-type and depleted strains are apparent, is shown in Fig. 5a.

Very strong effects are seen in some cases, as for instance in the case of the *pus4Δ* strain, where not only the Ψ55 modification is absent after a long incubation in the corresponding yeast extract, as expected, but the m$^1$A58 modification is absent as well. In addition, the incorporation of T54 is substantially hindered in the *pus4Δ* strain. Likewise, the incorporation of m$^1$A58 is much reduced in the *trm2Δ* strain (preventing T54 formation), which altogether suggest a regulatory circuit in the T-arm of tRNA$^{Phe}$. In this circuit of modification, Ψ55 has a positive effect on the introduction of T54 and m$^1$A58 by their respective enzymes (Supplementary Table 1), and T54 has a positive effect on the introduction of m$^1$A58 (Fig. 5b). Another strong effect is observed in the case of the *trm11Δ* strain (preventing m$^2$G10 formation), where the m$^2_2$G26 modification is apparent after ~14 h of incubation in the *trm11Δ* yeast extract, but is undetectable in wild-type and all other depleted strain extracts after ~24 h of incubation. Therefore, m$^2$G10 has a negative effect on the introduction of m$^2_2$G26 (Fig. 5b). Less drastic effects are also apparent in the different time-courses of the tRNA$^{Phe}$ modification pathway (Fig. 4a and Supplementary Figs. 2–8), and reveal more subtle interplay in the introduction of certain modifications. For instance, dihydrouridines introduced in the D-arm by Dus1 are seen to have a positive effect on the introduction of m$^1$A58 in the T-arm and a negative effect on the introduction of m$^2$G10 in the D-arm (Fig. 4a and Supplementary Fig. 2). Moreover, m$^2_2$G26 has a positive effect on the introduction of the nearby m$^7$G46 but also on the more distant m$^1$A58 in the T-arm (Fig. 4a and Supplementary Fig. 4). Our analysis has revealed a complex interplay between different modifications in the tRNA core, with the introduction of some modifications appearing to be strongly coupled (Fig. 5b). We next set out to analyze these modification circuits in vivo.

**Modification circuits in other yeast tRNAs.** The modification circuits identified here in yeast extracts are very clear, but we wanted to rule out the possibility that these interdependences could be the consequence of monitoring the modification pathways in yeast extracts, in which numerous cellular activities are present but many others are likely missing. In addition, we wondered whether the modification circuits uncovered here on tRNA$^{Phe}$ are specific to this tRNA or whether they are relevant to tRNA modifications in general. To address these two points, we undertook to quantify the modification content of total yeast tRNAs prepared from different yeast strains depleted of one specific modification enzyme at a time. The absolute quantification of the different modified nucleosides was performed by liquid-chromatography coupled with tandem mass spectrometry (LC–MS/MS) and compared with the modification content of wild-type yeast cultured under the same experimental conditions.

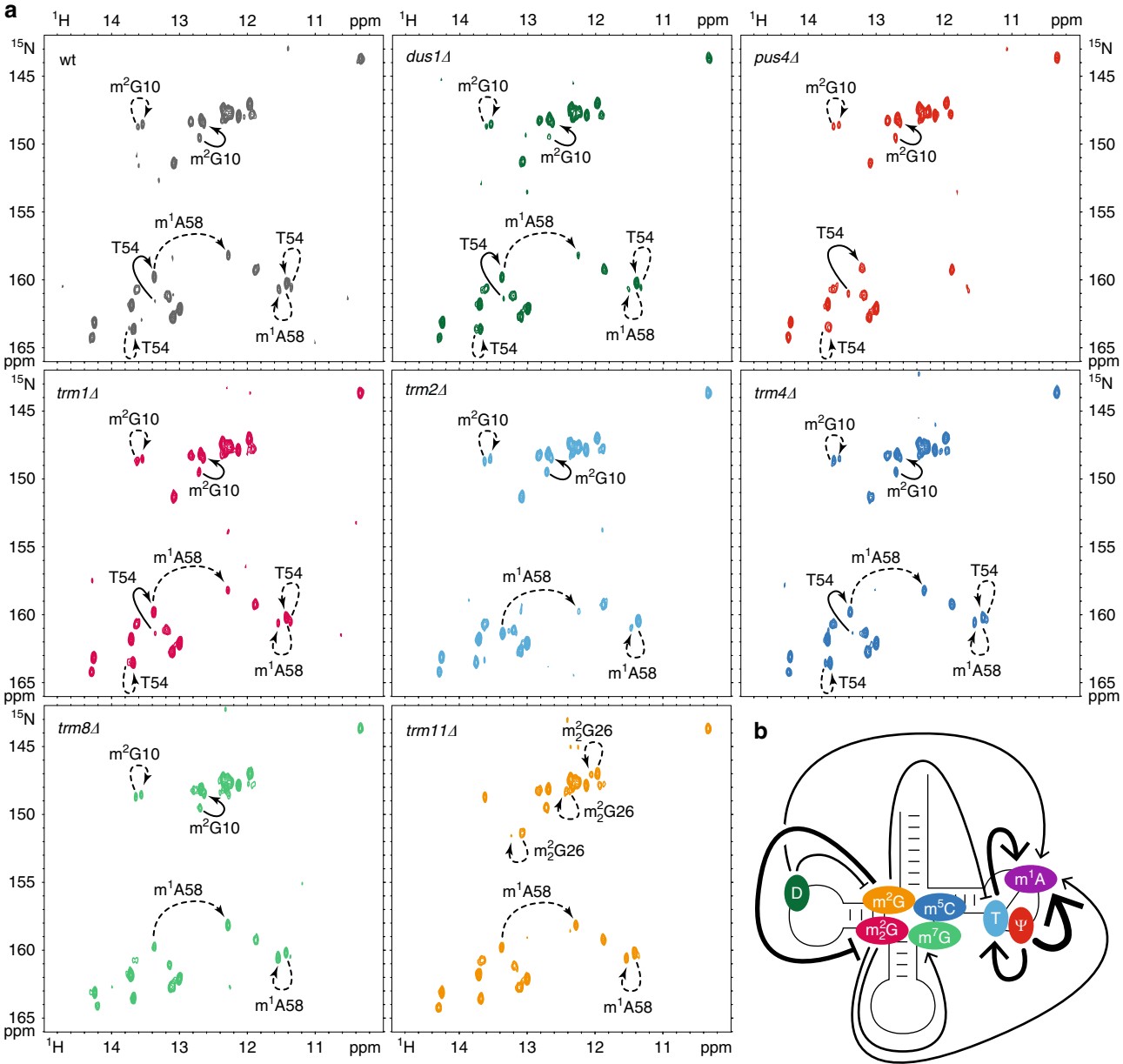

**Fig. 5** Complex modification circuits in yeast tRNA$^{Phe}$. **a** Imino ($^1$H,$^{15}$N) correlation spectra of a $^{15}$N-labeled tRNA$^{Phe}$ measured after 16 h of incubation at 30 °C in yeast extracts prepared from wild-type cells (top left spectrum) and in yeast extracts depleted of one modification enzyme at a time, namely *dus1Δ*, *pus4Δ*, *trm1Δ*, *trm2Δ*, *trm4Δ*, *trm8Δ*, and *trm11Δ*. Each NMR spectrum measurement spreads over a 2 h time period ($t = 16–18$ h). Complete series of time-resolved NMR monitoring of tRNA$^{Phe}$ maturation in the different depleted yeast extracts are given in Supplementary Figs. 2–8. Modifications occurring at this specific step in each extract are reported with continuous line arrows for direct effects, or dashed arrows for indirect effects. Each strain is associated with a defined color also used in panel **b**. **b** Schematic view of the modification circuits revealed by the NMR monitoring of tRNA$^{Phe}$ maturation in the different yeast extracts of panel **a**. Each modification is displayed on the cloverleaf structure with its associated color also used in Figs. 4 and 6. Arrows indicate stimulatory effects and blunted lines inhibitory effects. Thick and thin lines indicate strong and slight effects, respectively

Modification abundances were quantified by isotope dilution mass spectrometry[39] for the following modifications: D, Ψ, m$^2_2$G, T, m$^5$C, m$^7$G, m$^2$G, and m$^1$A in wild-type and depleted strains as used for the NMR monitoring in yeast extracts, namely *dus1Δ*, *pus4Δ*, *trm1Δ*, *trm2Δ*, *trm4Δ*, *trm8Δ*, and *trm11Δ* (Fig. 6). Their quantification in total tRNAs (Fig. 6a, b) reflects the sum of the individual modification changes in all tRNA species and therefore alterations specific to a single tRNA are most probably averaged out and masked in these measurements. Substantial changes in the tRNA modification contents of depleted strains therefore reflect changes that must be common to several tRNA species. Apart from the expected loss of modifications corresponding to

the depleted gene itself, significant alterations in the modification content of total tRNAs were measured in the *pus4Δ* and *trm2Δ* strains (Fig. 6a, b and Supplementary Fig. 9). In the *pus4Δ* strain, both the levels of T and m$^1$A are much reduced, whereas in the *trm2Δ* strain, the level of m$^1$A is slightly but significantly reduced (Fig. 6b). This shows that the effects we have observed with NMR in yeast extracts on tRNA$^{Phe}$ (namely that Ψ55 has a positive effect on the introduction of T54 and m$^1$A58, and T54 has a positive effect on the introduction of m$^1$A58) also occur in vivo and must be present in several tRNAs. Our measurements clearly establish the existence of a conserved modification circuit involving T54, Ψ55, and m$^1$A58 in the T-arm of tRNAs in yeast, but

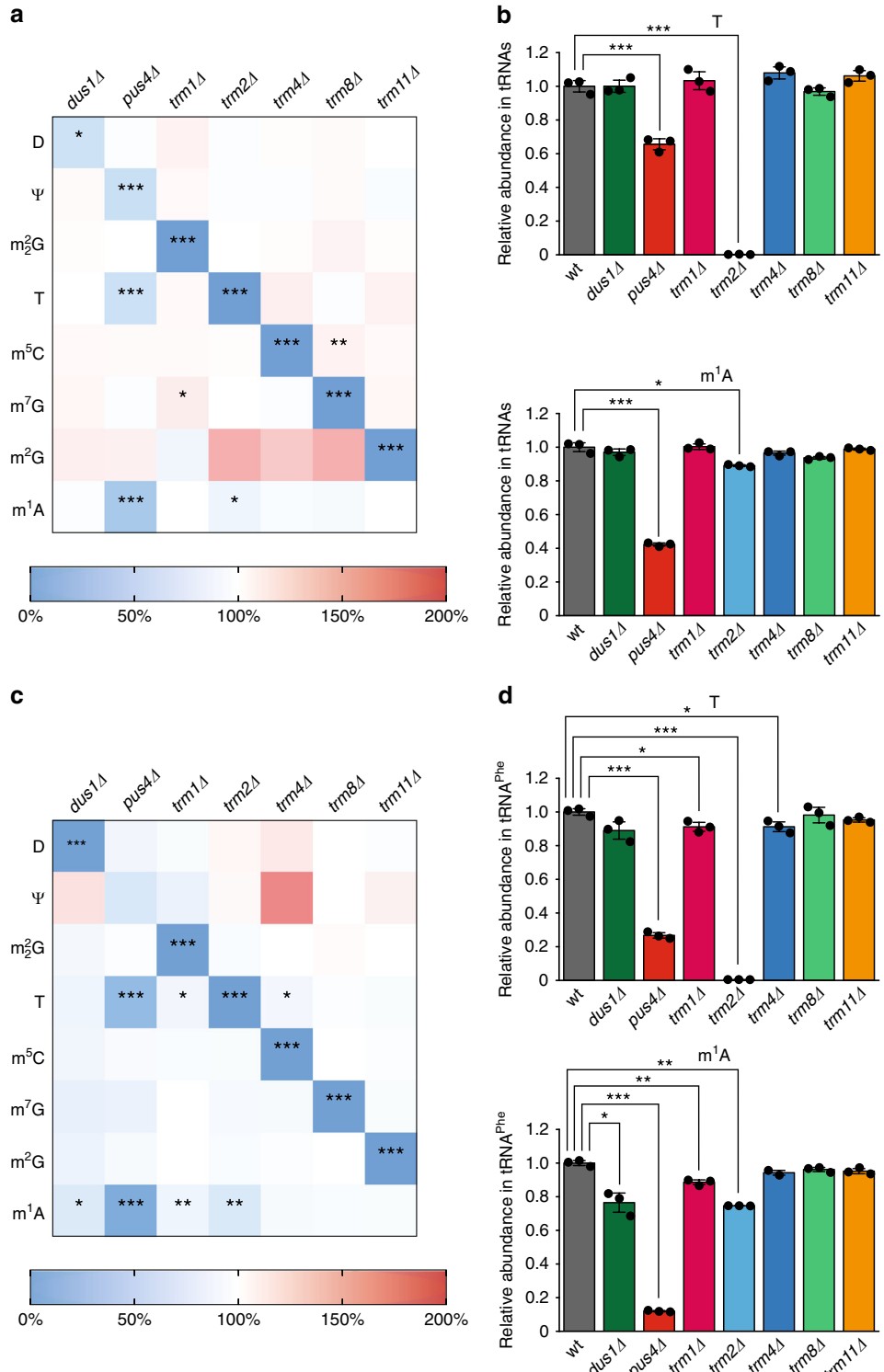

the extent to which a given tRNA is responsive to this circuit is likely to be tRNA-species-dependent (see the "Discussion" section). Apart from this modification circuit in the T-arm, other more subtle interdependences detected by NMR on tRNA$^{Phe}$ were not observed on total tRNAs (Figs. 5 and 6a, b), which led us to quantify the modification content of specifically purified tRNA$^{Phe}$ prepared from these same total tRNA samples. The number of modified nucleosides in tRNA$^{Phe}$ from wild-type yeast is shown in Supplementary Fig. 10, while the relative abundance is shown in Fig. 6c, d and Supplementary Fig. 11). Similarly to

what was observed for total tRNAs, the levels of T and m$^1$A in the purified tRNA$^{Phe}$ samples confirm interdependence of modifications in the T-arm of tRNA$^{Phe}$ (Fig. 6c, d and Supplementary Fig. 11). It is worth noting that the effects on the T and m$^1$A levels in the *pus4Δ* strain are even more pronounced for tRNA$^{Phe}$ than in total tRNAs, with almost no m$^1$A detected in this strain in tRNA$^{Phe}$ (Fig. 6b, d). In addition, dihydrouridines introduced by Dus1 in the D-arm and m$^2_2$G26 introduced by Trm1 have both a positive effect on the introduction of m$^1$A58 in the T-arm (Fig. 6d). These effects were also observed by NMR in yeast

**Fig. 6** Quantitative analysis of nucleoside modifications in yeast tRNAs with LC–MS/MS. **a** Heat map depicting the relative comparison of normalized modification levels in total yeast tRNAs prepared from depleted strains (*dus1Δ*, *pus4Δ*, *trm1Δ*, *trm2Δ*, *trm4Δ*, *trm8Δ*, and *trm11Δ*) using the wild-type levels as reference. The nucleotides quantified by LC–MS/MS are listed on the left side of the map. The scale bar indicates the fold change in modification levels compared with the wild-type strain (increased levels shown as red, no change as white, and decreased levels as blue). **b** Histograms showing the relative abundance of T and $m^1A$ modifications in total yeast tRNAs prepared from the depleted strains using the wild-type levels as reference. **c** Heat map depicting the relative comparison of normalized modification levels in specifically purified yeast $tRNA^{Phe}$ prepared from the depleted strains using the wild-type levels as reference. The scale bar indicates the fold change as in panel **a**. **d** Histograms showing the relative abundance of T and $m^1A$ modifications in specifically purified yeast $tRNA^{Phe}$ prepared from the depleted strains using the wild-type levels as reference. In panels **b** and **d**, black dots represents individual measurements, data heights represent the mean of the biological replicates. Error bars correspond to the s.d. In all panels, significant changes compared to wild type are reported as *** for $p < 0.001$, ** for $p < 0.01$ and * for $p < 0.05$, $n = 3$. All other changes are not statistically significant. Statistical analyses of the variations compared to the wild-type strain were performed using a two-sided Student's *t*-test. See also Supplementary Figs. 9 and 11 for similar analysis of all modified bases in total tRNA and purified $tRNA^{Phe}$. Modifications were quantified in three independent biological replicates, except for the modifications Ψ, $m^7G$, and $m^1A$ in the specifically purified $tRNA^{Phe}$ of the *trm4Δ* strain, for which $n = 2$. Source data are provided as a Source Data file

extracts (Fig. 5). Finally, our mass spectrometry (MS) measurements suggest other small effects that were not detected by NMR in yeast extracts, such as $m_2^2G26$ and $m^5C$ modifications, which both have a slight but significant positive effect on the introduction of T54 (Supplementary Fig. 11). Overall, the quantification of modifications performed here with MS demonstrates that interdependences between modifications observed by NMR in yeast extracts on $tRNA^{Phe}$ also occur in yeast in vivo and that some of these modification circuits are not limited to $tRNA^{Phe}$ but are common to several yeast tRNAs.

## Discussion

In this study, we developed an original methodology for monitoring the introduction of modified nucleotides in tRNAs using NMR in a time-resolved fashion. Our methodology enabled us to demonstrate that the modifications in yeast $tRNA^{Phe}$ are introduced in a particular order and that there are complex interplays between certain modifications. In particular, modifications T54, Ψ55, and $m^1A58$ in the T-arm are strongly interconnected in a circuit of modifications that drives the modification process along a defined route, namely Ψ55, then T54 and finally $m^1A58$. Our study has important implications for our understanding of the regulation of tRNA modifications in yeast. These aspects are discussed below.

By monitoring the introduction of modifications in tRNAs in a time-resolved fashion with NMR, our methodology enables the identification of sequential events and therefore reveals potential regulatory circuits in the system under study. If the introduction of a certain modification is influenced by the presence of prior modifications, this dependency indeed drives a defined order in the introduction of these modifications. Only a handful of modification circuits have been reported to date, most probably because their study remains difficult, since monitoring the maturation of tRNA in real-time at a single nucleotide level is technically challenging[18]. Most of the well-documented examples of modification circuits occur in the ACL and involve modifications at the frequently modified positions 34 and 37[1,4]. Although the reasons for which modification circuits exist in tRNAs are not known, it was recently proposed that the first modifications act as additional recognition elements for other modifying enzymes, which provides the mean for adding modifications with considerable variation in the ACL[14]. This theory is highly convincing for modifications in the ACL, but cannot account for the modification circuit identified here in the T-arm of yeast tRNAs. The three modifications involved are indeed highly conserved, with T54 and Ψ55 being present in all but the initiator tRNA in yeast (the specific case of $tRNA_i^{Met}$ is discussed below), and the $m^1A58$ modification being present in about two-thirds of yeast cytosolic tRNAs[1]. Similarly, other small effects uncovered here in $tRNA^{Phe}$, for instance the dihydrouridines in the D-arm and the $m_2^2G26$

modification, which have a positive influence on the introduction of the $m^1A58$ modification in the T-arm (Fig. 5 and Supplementary Fig. 11), cannot be explained by the need for an increased modification variability in the tRNA core, since these modifications are also highly conserved in yeast tRNAs. The modification circuits uncovered here in the tRNA core of yeast tRNAs are reminiscent of the circuits that have been reported in *Thermus thermophilus* tRNAs[40–42]. Although different in the exact nature of the connections between modifications, both the modification circuits revealed here in yeast and those found in *T. thermophilus* involve conserved modifications from the tRNA core and link modifications from several tRNA regions such as the T- and D-arms[42]. In the case of *T. thermophilus*, the modification circuits influence and regulate the levels of modifications in response to changes in external temperature, a mechanism that has been linked to an adaptation of protein synthesis to temperature change[43]. Whether the circuits identified here in yeast are part of a regulatory mechanism for adaptation to environmental changes is a stimulating idea that deserves further studies.

It is worth noting the self-coherence of the modification circuits reported here in yeast tRNAs. For instance, the branch D ⊣ $m_2G10$ ⊣ T54 → $m^1A58$ is consistent, regarding the influence of the dihydrouridine modifications on the $m^1A58$, with the direct branch D → $m^1A58$ (Fig. 5b). Similarly, the branch Ψ55 → T54 → $m^1A58$ is consistent with the direct branch Ψ55 → $m^1A58$ (Fig. 5b). It should be remembered that the type of global analysis that we carried out here with NMR or MS, cannot distinguish direct from indirect effects. The effects that appear in the form of a short-branch circuit might therefore be indirect effects arising from accumulated direct effects in a long-branch circuit with identical endpoints. To discriminate direct from indirect effects is an important point that could be addressed in vitro by performing careful enzymology with purified enzymes and tRNA substrates bearing or not pre-existing modifications. Among other techniques, this approach could be implemented with NMR using different types of labeling, e.g. $^{15}N$-labeling of the tRNA substrate and a similar experimental setup as presented here, or $^{13}C$-labeling of the transferable methyl of the SAM cofactor and a monitoring of the modification process with $^1H$–$^{13}C$ correlation spectra showing the incorporation of the methyl groups into the tRNA substrate. However, whether some of the effects observed here are direct or indirect does not radically change their fundamental quality: a connection exists between two modifications.

If one compares the effects of modifications observed with NMR on $tRNA^{Phe}$ (Fig. 5) and those measured by MS on total tRNAs (Fig. 6a and Supplementary Fig. 9), it seems reasonable to conclude that the identity of the tRNA has an influence on the conservation and the intensity of a given modification circuit. Apart from the effects of the Ψ55 → T54 → $m^1A58$ modification circuit in the T-arm, the other effects observed on $tRNA^{Phe}$ with

NMR are not apparent in the MS analysis of total tRNAs. This means that these particular effects are not found in all the different tRNAs but are averaged out and therefore are not detected in the total tRNA population. The observed differences are not derived from potential discrepancies between NMR and MS analysis, since the MS measurements on purified tRNA$^{Phe}$ are more similar to the NMR measurements than the MS data on total tRNAs (Figs. 5 and 6 and Supplementary Figs. 9 and 11). This conclusion is corroborated by the difference in the absolute quantification of modified nucleosides performed by MS on total tRNA and purified tRNA$^{Phe}$. As a comment, it is worth mentioning that the differences that exist between the NMR and MS measurements on purified tRNA$^{Phe}$ can have two different origins. First, they can reflect the fact that cell extracts do not fully recapture the enzymatic activities of intact cells, especially regarding protein localization (see below). Second, our approach examines the de novo synthesis of modifications on an unmodified tRNA with NMR and the steady-state levels of modifications in mature tRNAs with MS. Differences can therefore arise from the fact that the relatively weak effects on rates of modifications detected by NMR during de novo synthesis might not give rise to significant changes in the steady-state levels of modified nucleosides measured with MS. Concerning the role of tRNA identity, it has been proposed that some modifications might matter more in certain tRNA species than in others[44]. Our data are consistent with the extension of this idea to the connections between modifications, meaning that modification circuits may have more pronounced effects in certain tRNA species than in others. From a mechanistic point of view, this means that for a certain tRNA species, which is an intrinsically poor substrate of a given enzyme, decent modification activity is only achieved in the presence of a prior modification. Whereas for another tRNA species, an intrinsically good substrate of the same enzyme, the presence of a prior modification, although beneficial, has only a minor effect on the catalysis. In such a situation, the modification circuits would have a more pronounced effect on the first tRNA species than on the second one. In accordance with this idea, in the $\Psi$55–T54–m$^1$A58 modification circuit in the T-arm, which is certainly conserved in several tRNAs (Fig. 6b), the strength of the connections most likely depends on the identity of the tRNA. The effects observed in the pus4$\Delta$ and trm2$\Delta$ strains are indeed more pronounced on tRNA$^{Phe}$ than on total tRNAs (Fig. 6b, d), meaning that the effects of these circuits are necessarily less pronounced on other tRNA species. It is important to mention here that this reasoning can be safely applied in this particular case, since the m$^1$A58 and T54 modifications are the only source of m$^1$A and T in yeast tRNAs[1].

Although these extract-based NMR measurements may not accurately reflect the spatially controlled enzyme activities of intact cells, they nevertheless permit comparative assessments of global RNA modification activities and their changes upon depletion of a specific modification enzyme. In other words, even though our NMR approach might be biased by the lack of compartmentalization, by the use of a relatively high concentration of substrate tRNA (40 $\mu$M) that stretches the modification process over an artificial time span, by an intrinsic decay of enzymatic activities during incubation that is likely enzyme-dependent, and by the fact that the transcription is here dissociated from the modification process, it enables the identification of connections between modifications. In support of our methodology, most of the modification circuits identified in extracts with NMR have been corroborated by MS data on tRNAs purified from living yeast cells. The most striking difference between the NMR-derived circuits and the MS measurements concerns the strong negative influence of m$^2$G10 on the introduction of m$^2_2$G26 observed with NMR, a connection which is

absent from the MS data (Figs. 5 and 6). We believe that the significant inhibition observed in NMR describes a true reduction of the Trm1 activity in the presence of m$^2$G10, an effect that is perfectly compatible with the three-dimensional structure of yeast tRNA$^{Phe}$, in which m$^2$G10 and m$^2_2$G26 stack on one another (Supplementary Fig. 1)[38]. However, this situation might never arise in cells as a consequence of the subcellular localization of the different enzymes. Indeed, Trm11 (responsible for m$^2$G10) is a cytoplasmic protein whereas Trm1 (producing m$^2_2$G26) is a nuclear protein, and since the m$^2_2$G26 formation is insensitive to the presence of the intron, the m$^2_2$G26 modification most likely occurs before the primary export of tRNAs to the cytoplasm (Supplementary Table 1)[45–47]. The m$^2_2$G26 modification is thus introduced on tRNAs lacking the m$^2$G10 modification, and in the light of our data it is tempting to speculate that Trm11 is localized in the cytoplasm to avoid it inhibiting the m$^2_2$G26 modification. In another example, the subcellular location of Trm2 is unknown[8], but there is indirect evidence of an m$^5$U54-methylation activity in the nucleus, and Trm2 is therefore thought to be, at least partially, a nuclear protein[48–50]. Our data perfectly agree with a nuclear Trm2 protein. In the trm2$\Delta$ strain, the levels of m$^1$A, introduced at position 58 by the Trm6/Trm61 nuclear complex[51], are indeed slightly but significantly reduced (Fig. 6b, d). This means that the T54 modification has a slight positive effect on the introduction of m$^1$A58, an effect that is incompatible with a situation, where tRNAs would be modified first with m$^1$A58 in the nucleus by Trm6/Trm61 and modified by Trm2 in the cytoplasm after tRNA export.

In this study, we report a robust modification circuit in the T-arm of yeast tRNAs, with $\Psi$55 positively influencing the introduction of both T54 and m$^1$A58, and T54 positively influencing the introduction of m$^1$A58 (Figs. 5 and 6). This modification circuit was initially postulated based on the ordered modification process revealed by the time-resolved monitoring of tRNA$^{Phe}$ modifications in yeast extracts (Fig. 4). A survey of the tRNA$^{Phe}$ modification process in yeast extracts using radiolabelled nucleotides and 2D chromatography on thin-layer plates to identify modified nucleotides has been previously reported[28]. Several aspects of the reported kinetics are in accordance with our observations, with for instance the pseudouridine modifications being the fastest modification introduced, but the ordered modification process in the T-arm was not identified. The T54 modification is indeed introduced with no lag-phase, arguing for T54 being a modification that does not require any prior modification. Unfortunately the m$^1$A58 modification was not detected in their experimental conditions, preventing the examination of a potential ordered modification process involving this nucleotide[28]. We believe it is worth emphasizing that our methodology has the advantage of providing information on modifications within their sequence context, meaning that one can differentiate, for instance, the m$^5$C40 from the m$^5$C49 modification, or the different pseudouridine modifications, information that is absent when tRNAs are digested down to nucleotides before analysis. Since nucleotides 54, 55, and 58 are almost absolutely conserved as U54, U55, and A58 in tRNA genes, and since these positions are often modified to T54, $\Psi$55, and m$^1$A58 in the three domains of life[4,52], it would be interesting to understand whether the T54–$\Psi$55–m$^1$A58 modification circuit identified in yeast, is conserved in other organisms. E. coli lacks m$^1$A58 modifications in its tRNAs, but interestingly, the enzyme catalyzing the $\Psi$55 modification in E. coli (TruB) prefers to bind unmodified tRNA, whereas the enzyme catalyzing the T54 modification (TrmA) binds tRNA containing $\Psi$55 more strongly (Ute Kothe, personal communication). This suggests that the $\Psi$55 and T54 modifications are likely introduced in the same order in E. coli and yeast. In T. thermophilus, complex modification

circuits exist in the T-arm, including the $m^1A58$ modification, which is stimulated by T54[53]; the $s^2T54$ modification of T54, which is stimulated by $m^1A58$[54]; and the $s^2T54$ and $m^1A58$ modifications, which are negatively regulated by Ψ55[41]. Although the yeast and _T. thermophilus_ circuits have the T54 → $m^1A58$ branch in common, the two circuits are fairly different. First, in _T. thermophilus_, the $m^1A58$ modification is negatively regulated by Ψ55, whereas it is strongly favored by Ψ55 in yeast (Figs. 5 and 6). And second, the connections in _T. thermophilus_ lead to slight changes in the modification content of the analyzed tRNAs (±10% compared to wild-type), whereas the strong influence of Ψ55 on $m^1A58$ in yeast $tRNA^{Phe}$ produces large changes, with the $m^1A58$ level in the _pus4Δ_ strain decreasing to ~10% of the wild-type level (Fig. 6d). Altogether, even though some connections might be conserved in different species, potential modification circuits in the T-arm of tRNAs are likely organism-dependent.

Finally, it is worth mentioning that the _trm6/trm61_ genes, coding for the heterodimer catalyzing the $m^1A58$ modification, are among the few genes coding for tRNA modification enzymes that are essential in yeast[6]. It might seem counterintuitive at first sight that the deletion of the _pus4_ gene, which causes a dramatic decrease in the level of $m^1A58$ in tRNAs (Fig. 6), results in no detectable phenotype in yeast[55]. However, the essentiality of the $m^1A58$ modification has been studied in detail and was shown to correspond to an increased and deleterious instability of the initiator $tRNA_i^{Met}$ when lacking $m^1A58$[56,57]. The initiator $tRNA_i^{Met}$ is the only yeast cytoplasmic tRNA lacking both the T54 and Ψ55 modifications[1]. Yeast $tRNA_i^{Met}$ has a very peculiar T-loop sequence, and contains unmodified A54 and U55, which together with other features give rise to a particular tRNA elbow structure[58]. Overall, initiator $tRNA_i^{Met}$ has its own pathway of modification in the T-arm, which does not depend on the T54–Ψ55–$m^1A58$ modification circuit identified here, and the level of $m^1A58$ in $tRNA_i^{Met}$ is therefore most likely unaffected in the _pus4Δ_ and _trm2Δ_ strains.

Overall, our work establishes NMR spectroscopy as an enlightening technique to analyze tRNA modification pathways. Our innovative methodology indeed reports on the sequential order of modification events and provides valuable information on the regulatory circuits in tRNAs. We expect that our NMR-based methodology will be applicable to investigate several aspects of tRNA maturation and RNA modifications in general, such as the dynamic adaptation of RNA modifications in response to environmental changes.

## Methods

**Yeast strains**. Yeast strains used in this study are listed in Supplementary Table 2. The wild-type _S. cerevisiae_ BY4741 strain and the YKO collection kanMX strains carrying deletions of the genes for modification enzymes (YML080w, YNL292w, YDR120c, YKR056w, YBL024w, YDL201w, and YOL124c) were obtained from Euroscarf and used for tRNA preparations for MS analysis.

The proteinase-deficient _S. cerevisiae_ strain c13-ABYS-86[59] was used for the preparation of yeast extracts used in NMR experiments. Chromosomal deletion of genes coding for modification enzymes (YML080w, YNL292w, YDR120c, YKR056w, YBL024w, YDL201w, and YOL124c) was carried out by homologous recombination in the c13-ABYS-86 genetic background. The kanamycin-resistance cassettes kanMX were amplified by high-fidelity PCR (Phusion, ThermoFisher) from the appropriate YKO collection kanMX strains (Euroscarf) followed by transformation of the DNA into the c13-ABYS-86 strain and selection on plates containing G418 at 300 μg/mL. All strain constructions were verified by PCR using appropriate oligonucleotides (listed in Supplementary Table 3).

**$tRNA^{Phe}$ samples for NMR**. Unmodified yeast $tRNA^{Phe}$ was prepared by standard in vitro transcription with T7 polymerase with unlabeled NTPs (Jena Bioscience) for unlabeled samples or $^{15}N$-labeled UTP and GTP (Eurisotop) and unlabeled ATP and CTP (Jena Bioscience) for $^{15}N$-[U/G]-labeled samples. The DNA template and T7 promotor primer were purchased from Eurogentec. The transcript was purified by ion exchange chromatography (MonoQ, GE Healthcare) under native conditions, dialyzed extensively against Na-phosphate pH 6.5 1 mM, and refolded by heating at 95 °C for 5 min and cooling down slowly at room

temperature. Buffer was added to place the $tRNA^{Phe}$ in the NMR buffer (Na-phosphate pH 6.5 10 mM, $MgCl_2$ 10 mM), and the sample was concentrated to ~1.5-2.0 mM using Amicon 10,000 MWCO (Millipore).

The yeast $^{15}N$-labeled $tRNA^{Phe}$ sample produced in and purified from _E. coli_ was prepared following previously published procedures[60]. Briefly, the gene coding yeast $tRNA^{Phe}$ was cloned in the pBSTNAV vector (Addgene ID 45801), expressed in LB for unlabeled sample preparation or $^{15}N$-labeled Spectra-9 medium (Eurisotop) for $^{15}N$-labeled sample preparation. After standard procedures of phenol extraction of soluble RNAs, yeast $tRNA^{Phe}$ was purified by ion exchange chromatography under native conditions, dialyzed against the NMR buffer and concentrated to ~0.5-1.0 mM.

Unlabeled matured yeast $tRNA^{Phe}$ was purchased from Sigma, resuspended in and dialyzed against the NMR buffer and concentrated to 0.8 mM.

**Yeast extract preparation**. Yeasts (c13-ABYS-86 strain) were grown in YEPD medium (1% (w/v) yeast extract, 2% (w/v) peptone, 2% (w/v) glucose) for 24 h at 30 °C and harvested by centrifugation. Pellets were stored at −80 °C until further use. Pellets were unfrozen and resuspended in their same weight of lysis buffer ($Na_2HPO_4$/$KH_2PO_4$ pH 7.0 25 mM, $MgCl_2$ 10 mM, EDTA 0.1 mM) complemented with 2 mM dithiothreitol (DTT), 1 mM phenylmethylsulphonyl fluoride (PMSF), 1 mM benza-midine, and 1 μg/mL each of leupeptin, pepstatin, antipain, and chymostatin. Cells were rapidly frozen and lysed in an Eaton pressure chamber[61] at 30,000 psi in a hydraulic press. The homogenate was centrifuged at 30,000 × g for 1 h at 8 °C to remove cellular debris. The supernatant was further centrifuged at 100,000 × g for 1 h at 8 °C. The resulting supernatant (15–20 mg/mL of proteins) was quickly aliquoted and frozen in liquid nitrogen. Aliquots were stored at −80 °C until used. Yeast extracts from the depleted strains were prepared following the same procedure.

**NMR spectroscopy**. All in vitro NMR spectra of yeast $tRNA^{Phe}$ were measured at either 38 or 30 °C on Bruker AVIII-HD 600 MHz and AVIII-HD 700 MHz spectrometers (equipped with TCI 5-mm cryoprobes) with 5-mm Shigemi tubes. Imino resonances of the $tRNA^{Phe}$ samples were assigned using 2D jump-and-return-echo ($^1H,^1H$)-NOESY[62,63], 2D ($^1H,^{15}N$)-BEST-TROSY[30], and standard 2D ($^1H,^{15}N$)-HSQC experiments measured in the NMR buffer ($NaH_2PO_4$/$Na_2HPO_4$ pH 6.5 10 mM, $MgCl_2$ 10 mM) supplemented with $D_2O$ 5% (v/v). NMR spectra in yeast extracts were measured at 30 °C on Bruker AVIII-HD 700 MHz spectrometer with 5-mm Shigemi tubes. Unmodified $^{15}N$-[U/G]-labeled $tRNA^{Phe}$ were prepared at 40 μM in yeast extracts (final concentration of 11 mg/mL of proteins) supplemented with $NaH_2PO_4$/$K_2HPO_4$ pH 7.5 150 mM, $NH_4Cl$ 5 mM, $MgCl_2$ 5 mM, DTT 2 mM, EDTA 0.1 mM, SAM 4 mM, ATP 4 mM, NADPH 4 mM, and $D_2O$ 5% (v/v)[64]. This concentration of $tRNA^{Phe}$ (40 μM) was chosen to achieve sufficient SNR in NMR measurements while seeking to approach cellular tRNA concentrations (as a comparison, the concentration of total tRNAs have been estimated to 100–200 μM in yeast and 200–350 μM in _E. coli_, with typical concentrations of individual tRNAs of 2–15 μM[65,66]). For monitoring the maturation of $tRNA^{Phe}$ in yeast extract, each 2D ($^1H,^{15}N$)-BEST-TROSY experiment was measured with a recycling delay of 200 ms, a SW($^{15}N$) of 26 ppm, 96 increments for a total experimental time of 120 min. A reference 2D ($^1H,^{15}N$)-BEST-TROSY spectrum of unmodified $tRNA^{Phe}$ was also measured at 303 K in a buffer containing $NaH_2PO_4$/$K_2HPO_4$ pH 6.75 100 mM, $MgCl_2$ 5 mM, ATP 4 mM, NADPH 4 mM and $D_2O$ 5% (v/v)[64]. The data were processed using TOPSPIN 3.5 (Bruker) and analyzed with Sparky (http://www.cgl.ucsf.edu/home/sparky/).

**Total tRNA samples from yeast for mass spectrometry**. Total tRNA from _S. cerevisiae_ BY4741 wild-type or mutant strains used for mass spectrometry analysis were prepared following previously published procedures[67]. For each strain, all cultures and tRNA preparations were performed in triplicate for statistical analysis. Briefly, the protocol was adapted as follows. Yeast cells were grown in YEPD medium at 30 °C. Yeasts were collected in logarithmic growth phase and washed with water. Pellets were resuspended in one volume of NaCl 150 mM and mixed with two volumes of water-saturated phenol. After mild shaking at room temperature for 30 min, one volume of chloroform was added and the mixture was vortexed for 15 min. The water and phenol phases were separated by centrifugation for 20 min at 8000 × g at 4 °C. RNAs were precipitated from the aqueous phase for 2 h at −20 °C by adding 2.5 volumes of cold ethanol and 0.1 volume of potassium acetate 2 M. RNAs were pelleted by centrifugation at 10,000 × g for 15 min at 4 °C. Dry pellets were resuspended in 400 μL of Tris–HCl pH 8.0 2 M and tRNAs were incubated for 90 min at 37 °C for aminoacyl-tRNA deacylation. RNAs were precipitated with 2.5 volumes of cold ethanol at −20 °C. Pellets were resuspended in 100 μL of Li/K-acetate buffer (lithium acetate 2 M, potassium acetate 0.1 M pH 5.0). After incubation at 4 °C for 20 min, most of the insoluble ribosomal RNAs were eliminated by centrifugation. Soluble tRNAs in the supernatant were recovered by precipitation with 2.5 volumes of cold ethanol. Finally, tRNAs were resuspended in Tris–HCl pH 7.5 1 mM, magnesium acetate 10 mM, and precipitated with 0.1 volume of ammonium acetate 5 M and three volumes of cold ethanol for 2 h at −20 °C. After centrifugation, tRNAs were dissolved in RNase-free water (Thermofisher). The total tRNA samples prepared with this procedure contains small contaminations of other small RNAs < 200 nts, mainly 5S and 5.8S rRNAs (both the 5S and 5.8S rRNA contain a single pseudouridine in their

sequence and no other modifications). Using size exclusion chromatography (SEC), tRNAs were estimated to account for ~80% of the small RNAs in these samples.

**Specific isolation of yeast tRNA$^{Phe}$.** Yeast tRNA$^{Phe}$ was isolated from ~1 µg total tRNA samples (RNAs < 200 nts) with a first step of SEC following previously published procedures[39,68], and a subsequent purification using T1 Dynabeads (Thermo Fisher Scientific, Product no. 65801D) and a DNA probe specific to tRNA$^{Phe}$ ([Btn]AAATGGTGCGAATTCTGTGGATCGAACACAGGACCTCCAG ATCTTC, Sigma-Aldrich, Munich, Germany) as previously reported[69].

**Digestion of tRNAs to nucleosides.** Total tRNA samples (300 ng for each sample) and the purified tRNA$^{Phe}$ samples were digested to single nucleosides for 2 h at 37 °C with alkaline phosphatase (0.2 U, Sigma-Aldrich, St. Louis, MO, USA), Phosphodiesterase I (0.02 U, VWR, Radnor, PA, USA), and Benzonase (0.2 U) in a buffer containing Tris–HCl pH 8.0 5 mM and MgCl$_2$ 1 mM. Tetrahydrouridine (THU, 0.5 µg from Merck), butylated hydroxytoluene (BHT, 1 µM), and Pentostatin (0.1 µg) were also added to protect modifications. Afterwards samples were filtered through multi-well plates (Pall Corporation, 10 kDa MWCO) at 4 °C for 30 min at 3000 × $g$ to remove digestive enzymes. Stable isotope-labeled internal standard (SILIS, 0.1 volume of 10X solution) from yeast was added for absolute quantification[39].

**Mass spectrometry.** For quantification, an Agilent 1290 Infinity II equipped with a DAD combined with an Agilent Technologies G6470A Triple Quad system and electro-spray ionization (ESI-MS, Agilent Jetstream) was used. Operating parameters were as follows: positive ion mode, skimmer voltage 15 V, cell accelerator voltage 5 V, N$_2$ gas temperature 230 °C, and N$_2$ gas flow 6 L/min, sheath gas (N$_2$) temperature 400 °C with a flow of 12 L/min, capillary voltage of 2500 V, nozzle voltage of 0 V, and the Nebulizer at 40 psi. The instrument was operated in dynamic MRM mode (individual mass spectrometric parameters for the nucleosides are given in Supplementary Table 4. The mobile phases were: A as 5 mM NH$_4$OAc (≥99%, HiPerSolv CHROMANORM®, VWR) aqueous brought to pH = 5.6 with glacial acetic acid (≥99%, HiPerSolv CHROMANORM®, VWR) and B as pure acetonitrile (Roth, LC–MS grade, purity ≥ 99.95%). A Synergi Fusion-RP column (Phenomenex®, Torrance, CA, USA; Synergi® 2.5 µm Fusion-RP 100 Å, 150 × 2.0 mm) at 35 °C and a flow rate of 0.35 mL/min was used. The gradient began with 100% A for 1 min, increased to 10% B by 5 min, and to 40% B by 7 min. The column was flushed with 40% B for 1 min and returned to starting conditions to 100% A by 8.5 min followed by re-equilibration at 100% A for 2.5 additional minutes.

For calibration, synthetic nucleosides were weighed and dissolved to a stock concentration of 1–10 mM. Calibration solutions ranging from 0.25 to 100 pmol for each canonical nucleoside and from 0.0125 to 5 pmol for each modified nucleoside were prepared in water ($D$, $\Psi$ = 0.025–10 pmol). The calibration solutions were mixed with the yeast SILIS and analyzed by LC–MS/MS. The value of each integrated peak area of the nucleoside was divided through the respective SILIS area. The linear regression for each nucleoside's normalized signal/ concentration plot gives the relative response factor for nucleosides (rRFN)[39]. The data were analyzed by the Quantitative and MassHunter Software from Agilent. Finally, the absolute amounts of the modifications were referenced to the absolute amounts of summed canonical nucleosides. The number of modifications per tRNA$^{Phe}$ was calculated by determining the amount of injected tRNA from the signal of canonicals and the number of canonicals from the tRNA$^{Phe}$ sequence. Statistical analyses of the variations compared to the wild-type strain were performed using a two-sided Student's $t$-test.

**Reporting summary.** Further information on research design is available in the Nature Research Reporting Summary linked to this article.

## Data availability

A reporting summary for this Article is available as a Supplementary Information file. The source data underlying Fig. 6 and Supplementary Figs. 9–11 are provided as a Source Data file. All data is available from the corresponding author upon reasonable request.

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

## Acknowledgements

The authors are grateful to Henri Grosjean for stimulating discussions about RNA modifications, Valérie Heurgué-Hamard (IBPC), Josette Banroques (IBPC), Simon Lebaron (CBI), and Sylvie Auxilien (I2BC) for yeast strains, protocols and helpful discussions about yeast handling, Bruno Sargueil for guidance regarding the Eaton press implementation, and Jacqueline Plumbridge (IBPC) for careful reading of the manuscript. This work was supported by grants from the CNRS, the ANR NMR-VitAmin (ANR-14-CE09-0012), the Labex DYNAMO (ANR-11-LABX-0011), the Equipex Cacsice (ANR-11-EQPX-0008) and the SESAME Île-de-France. P.B. and C.T. also acknowledge access to the NMR infrastructure of the Paris Descartes University that is supported by grants from the Région Île-de-France, the European Union (FEDER) and the Paris Descartes University. M.H. and S.K. are grateful for funding from the Deutsche Forschungsgemeinschaft (KE1943/3-1 and KE1943/4-1).

## Author contributions

C.T. conceived the initial project; P.B. and C.T. refined the project and designed the experiments; P.B. and M.C. prepared tRNA samples for NMR studies; P.B., A.G., and M.C. performed gene deletions in yeast and prepared yeast extracts for NMR studies; P.B. measured and analyzed NMR spectra; A.G. prepared total tRNA samples for MS; M.H. isolated tRNAPhe and digested tRNAs for MS; M.H. and S.K. measured and analyzed MS data; P.B. wrote the manuscript; all authors discussed the results, critically reviewed the manuscript and approved the final version.

## Additional information

**Competing interests:** The authors declare no competing interests.

