## [Peer Review File · Nature Communications]

Reviewers' Comments:

Reviewer #1:

Remarks to the Author:

In this work, Tisne and co-workers report on an NMR and MS based approach to identify modifications circuits in tRNA maturation. Using a real time NMR assay the chronology of the maturation process was addressed. For example a strong hierarchy was found for the T54, Y55 and m1A58 modification in the T-arm of yeast tRNAPhe. Further using MS the NMR results were put into context with the modification process in living cells.

The amount of presented experimental data is impressive. The data is further presented in a clear manner and interpreted carefully.

Thus, the work is certainly suitable for publication in Nature Communications and the results give a deeper understanding how the fundamental process of tRNA maturation/modification works.

I have a few minor points that should be addressed before publishing:

1. Did the authors consider to use ¹³C labeled SAM cofactor to monitor the built-up of the m1A, m5C, m7G, Cm, Gm, m22G, m2G modifications using the SOFAST HMQC approach originally developed for proteins?
2. Did the authors consider to use ¹H-¹³C detection using ¹³C labeled unmodified tRNA. This should give a significant increase in reporter nuclei number and the indirect effects should be directly detectable on the residue that is modified.
3. Did the authors try to use the D imino proton signals (at approx. 10 ppm ¹H and 150 ppm ¹⁵N) to detect the U to D transformation. This signal although not part of a base pairing interaction is observable at 25°C due to the pKa value of 12 of H3.
4. The discussion section ends quite abruptly. The authors could add a short perspective which other open questions can be addressed with the novel approach described in their work.

Reviewer #2:

Remarks to the Author:

The manuscript by Barraud et al entitled: Time-resolved NMR monitoring of tRNA maturation reports on the time-resolved monitoring of RNA modification of tRNA transcripts in protein extracts from yeast in vitro. The topic is very attractive, and the results very relevant and of significant value to the community. In addition, the description of the technique developed here, is of particular impact for future studies. Overall, my opinion is very positive. Indeed, if the impact of the respective work in the protein posttranslational modification field is any reference, the present manuscript may well become a milestone paper in the field.

The authors use ¹⁵N labeled, in vitro transcribed tRNAPhe as a model, recording posttranscriptional modification events "in real time" in the NMR instrument. Although the authors consider the concentration of 40 micromolar tRNA as "low", this needs to be discussed in a bit more details. For an NMR spectroscopist, that concentration is certainly low, but it is clearly vastly above physiological concentration. Clearly, a tRNA can be expected to be modified to completion a time span corresponding roughly to the doubling time of a yeast cell..... Accordingly, it is clear that the time span during which modification events are observed by NMR is artificially stretched, and presumably, the high amount of tRNA substrate is the reason. This does, in my view, not present a detrimental effect to the performance of the technique, but the results need to be interpreted with a bit more care than is apparent in the present manuscript. For example when the authors suggest fast and slow enzymes/activities, these might be caused by differential decay of the corresponding activities during the 24h period. Thus, can the authors provide data on the relative enzymatic activities in the yeast extracts at t=0 and t=24h? The authors were clearly aware of some of the potential caveats, and have therefore validated their conclusions by a reverse genetics approach, which, in my opinion, renders the manuscript very solid.

I hence emphatically recommend publication (with minor adjustments of the discussion as indicated).

Miro point: please straighten out the logic of the T54-m158 discussion at the bottom of page 11. The current explanation is very convoluted, and certain passages are unclear.

Reviewer #3:

Remarks to the Author:

A growing theme in the RNA modification field is the idea that many modifications may not occur independent of each other but rather follow specific routes, whereby a pioneer modification may be important for a subsequent one. Implicit in such a cascade is the high potential for regulation. However, despite much progress in improving detection methods for various modifications, it is still technically challenging to map modifications to specific positions in a tRNA molecule and indeed mass spectrometry is the golden standard. In the present study, Barraud et al. implement the use of time-resolved NMR to monitor the temporal order of modifications in tRNA. Although the work is carefully performed and it has its merits, there are major issues with the work, which in my opinion limits the approach as applied here.

1. The use of NMR to analyze modifications is not necessarily new, for instance follow the work of Paul Agris and independently Darrel Davis' groups through the years. Here, the new thing is the use of time-resolved NMR, clearly the authors have successfully provided positional information about various modifications, however, the limitations are not from the detection itself, but from the idea that mixing an isotopically labeled tRNA, which I assume is folded somehow, into extracts will accurately say anything about the temporal nature of modifications. The very fact that the tRNA may or may not be folded argues against the reality of their temporal measurements and what can really be gleaned from those measurements. For instance, how do we know that the order established here is really a reflection of the modification pathway and not rather the struggle for a certain enzyme that may act early on the pathway to modify a "fully" folded transcript. Here, a good case can be made with m²G¹⁰, since methylations at the 9 position are important for tRNA folding, it stands to reason that such methylations may occur early in the modification/folding pathway and there is evidence that it occurs co-transcriptionally. Thus it is impossible to think that pseudouridine 55 will occur before G⁹ methylations. Perhaps highlighting the shortcomings of this approach.

2. Another major issue deals with a number of extrapolations made with the analysis and conclusions about the results that are either not granted by the study or are simply factually wrong. For example, several times the authors mention an enzyme being "less active" or "more active" without providing any evidence of the specific activity of the enzyme in question, or any measurement of enzymatic rates. Along these lines, on page 12 the statement is made that "it is difficult to discriminate between authentic dependence and a sequential order..." this is simply not correct. Careful analysis by mechanistic enzymology can easily tease out reaction order and the enzyme mechanism.

3. Further limitations of the system are provided by the fact that their model substrate, tRNA^{Phe} has yW at position 37. The forcible precursor to yW is the methylated nucleotide m¹G³⁷, which happens independent of the other wybutosine enzymes and yet no trace of m¹G³⁷ is shown. It is not clear why this discrepancy.

So overall, despite the fact that I think time-resolved NMR to study modifications is a promising idea, the technique for the purpose proposed and discussed here is way too premature. Clear advantages over mass spectrometry are not so obvious.

Manuscript: NCOMMS-19-10130

Manuscript Title: Time-resolved NMR monitoring of tRNA maturation

Pierre Barraud, Alexandre Gato, Matthias Heiss, Marjorie Catala, Stefanie Kellner & Carine Tisné

Response to the reviewers' comments

We thank the three reviewers for their positive and constructive comments. We are happy to submit a revised version of our manuscript and hope that it is now suitable for publication in *Nature Communications*. Below we address each of the reviewers' comments:

Reviewer #1

In this work, Tisné and co-workers report on an NMR and MS based approach to identify modifications circuits in tRNA maturation. Using a real time NMR assay the chronology of the maturation process was addressed. For example a strong hierarchy was found for the T54, Y55 and m1A58 modification in the T-arm of yeast tRNAPhe. Further using MS the NMR results were put into context with the modification process in living cells. The amount of presented experimental data is impressive. The data is further presented in a clear manner and interpreted carefully. Thus, the work is certainly suitable for publication in Nature Communications and the results give a deeper understanding how the fundamental process of tRNA maturation/modification works.

We appreciate the positive feedback on our work.

I have a few minor points that should be addressed before publishing:

1. Did the authors consider to use ^{13}C labeled SAM cofactor to monitor the built-up of the m1A, m5C, m7G, Cm, Gm, m22G, m2G modifications using the SOFAST HMQC approach originally developed for proteins?

We thank the reviewer for this suggestion. This is indeed a direction we have considered to follow. So far, however, it has not been so straightforward to implement and we did not obtain valuable insights on tRNA maturation following this idea. Several aspects need indeed to be taken into consideration and explain the difficulties we have faced. First, strong undesirable signals are observed in the corresponding region in (^1H , ^{13}C) correlation spectra in cellular extracts. Small metabolites at 1-10 mM concentration give indeed already intense signals even at the natural abundance of ^{13}C and complicate the identification of small amount of modified tRNA at concentrations below and around 40 μM . Second, the SAM cofactor is the almost universal methyl donor cofactor, which is used by a very large number of enzymes. The ^{13}C -labelled methyl group is therefore not only incorporated into the tRNA of interest but also likely to all kind of substrates of SAM-dependent methylases (small molecules, proteins, RNAs...). The use of ^{15}N -labelling has the clear advantage that no background exist in this region of the (^1H , ^{15}N) correlation spectra in cellular extracts and that the tRNA of interest remains clearly the only ^{15}N -labelled molecule during the incubation. However, we believe that the ^{13}C -labelled SAM approach suggested by this reviewer is very likely to turn successful if implemented in a purely *in vitro* set-up with purified modification enzymes. This strategy is perfectly in line with the point 2 of reviewer #3 (see below), and we have added few sentences to discuss these aspects together page 18-19: “*To discriminate direct from indirect effects is an important point that could be addressed in vitro by performing careful enzymology with purified enzymes and tRNA substrates bearing or not pre-*

existing modifications. Among other techniques, this approach could be implemented with NMR using different types of labelling, e.g. ^{15}N -labelling of the tRNA substrate and a similar experimental setup as presented here, or ^{13}C -labelling of the transferable methyl of the SAM cofactor and a monitoring of the modification process with ^1H - ^{13}C correlation spectra showing the incorporation of the methyl groups into the tRNA substrate.”

2. Did the authors consider to use ^1H - ^{13}C detection using ^{13}C labeled unmodified tRNA. This should give a significant increase in reporter nuclei number and the indirect effects should be directly detectable on the residue that is modified.

Chemical shift assignment of uniformly (^{15}N , ^{13}C)-labelled RNAs above 50 nucleotides is technically very challenging. For tRNAs that are about 76 nucleotide-long, ^{13}C chemical shift assignments of the aromatics and of some sugar protons could however be achieved for instance by preparing multiple samples with nucleotide-specific labelling associated with partial deuteration of nucleotides (several strategies can be proposed, see for instance our recent review on these aspects Asadi-Atoi P. et al. *Biol Chem* (2019), *in press*). We did not yet follow this approach, and we thus thank this reviewer for the suggestion, which could indeed provide valuable information on modifications introduced on Ade and Cyt, as well as on modifications that are further away from stabilized imino-groups in stem regions, namely the anticodon loop.

3. Did the authors try to use the D imino proton signals (at approx. 10 ppm ^1H and 150 ppm ^{15}N) to detect the U to D transformation. This signal although not part of a base pairing interaction is observable at 25°C due to the pKa value of 12 of H3.

We thank the reviewer for this suggestion. This could have been indeed a very good mean to detect the U to D transformation. However, in our spectra we do not detect signals that could correspond to iminos of D (at approx. 10 ppm ^1H and 150 ppm ^{15}N), neither in the spectra measured *in vitro* (Figure 3b) nor in the series of spectra measured in yeast extracts. Following this reviewer idea, we could nevertheless try to optimize the measurement conditions for the detection of the iminos of D. We could first of all optimize the temperature and make measurements at 25°C as suggested, since our measurements were made at 38°C for the *in vitro* spectra and at 30°C for the spectra in yeast extracts. We will try to implement this in subsequent studies.

4. The discussion section ends quite abruptly. The authors could add a short perspective which other open questions can be addressed with the novel approach described in their work.

As suggested, we have placed a very short description of general perspectives at the end of the discussion section page 24.

Reviewer #2

The manuscript by Barraud et al entitled: Time-resolved NMR monitoring of tRNA maturation reports on the time-resolved monitoring of RNA modification of tRNA transcripts in protein extracts from yeast *in vitro*. The topic is very attractive, and the results very relevant and of significant value to the community. In addition, the description of the technique developed here, is of particular impact for future studies. Overall, my opinion is very positive. Indeed, if the impact of the respective work in the protein posttranslational modification field is any reference, the present manuscript may well become a milestone paper in the field.

We appreciate the positive and encouraging feedback on our work.

The authors use ^{15}N labeled, *in vitro* transcribed tRNA^{Phe} as a model, recording posttranscriptional modification events “in real time” in the NMR instrument. Although the authors consider the concentration of 40 micromolar tRNA as “low”, this needs to be discussed in a bit more details. For an NMR spectroscopist, that concentration is certainly low, but it is clearly vastly above physiological concentration. Clearly, a tRNA can be expected to be modified to completion a time span corresponding roughly to the doubling time of a yeast cell.... Accordingly, it is clear that the time span during which modification events are observed by NMR is artificially stretched, and presumably, the high amount of tRNA substrate is the reason. This does, in my view, not present a detrimental effect to the performance of the technique, but the results need to be interpreted with a bit more care than is apparent in the present manuscript. For example when the authors suggest fast and slow enzymes/activities, these might be caused by differential decay of the corresponding activities during the 24h period. Thus, can the authors provide data on the relative enzymatic activities in the yeast extracts at $t=0$ and $t=24\text{h}$? The authors were clearly aware of some of the potential caveats, and have therefore validated their conclusions by a reverse genetics approach, which, in my opinion, renders the manuscript very solid. I hence emphatically recommend publication (with minor adjustments of the discussion as indicated).

There are several points here.

First, we agree with this reviewer that the term “low concentration” is not appropriate. It is worth mentioning that intra-cellular concentration of total tRNAs have been estimated at 200-350 μM in *E. coli* depending on the growth rate, with typical concentrations of individual tRNA species ranging from 2 to 15 μM (Dong H. et al. J Mol Biol (1996) 260, 649–663). Similar concentrations have been estimated in yeast for total tRNAs (Waldron C. et al. J Bacteriol. (1975) 122, 855–865). But unfortunately, we could not find information on individual tRNA species in yeast.

The concentration we use for our NMR experiments in yeast extracts is certainly above physiological concentration (about 10x more than individual tRNAs), which indeed can in part explain the artificially stretched time span of tRNA modifications in our assays. We believe the lack of localization and probably the lack of controlled co-localization of enzymes and substrates is another reason explaining the apparently slow incorporation of modification in our assays.

Second, it is absolutely possible that the different enzymes present in the extracts have different intrinsic stabilities, with certain enzymes remaining fully active during the entire incubation, while others might lose activity for long incubation time. We have not tested this systematically, since we have instead set out to circumvent the potential limitations with a more global approach. Indeed, as pointed out by this reviewer, even though our system might be biased by the above mentioned points, the reverse genetic approach allows to put these potential caveats aside and enables the identification of true connections between modifications.

Overall, in order to follow this reviewer suggestion, as well as that of reviewer #3 (see answer to point 1 below), we have expanded a sentence in the discussion page 20-21, which now reads as: “[...] even though our NMR approach might be biased by the lack of compartmentalisation, by the use of a relatively high concentration of substrate tRNA (40 μM) that stretches the modification process over an artificial time span, by an intrinsic decay of enzymatic activities during incubation that is likely enzyme-dependent, and by the fact that the transcription is here dissociated from the modification process, it enables the identification of connections between modifications.”

In addition, we have removed the term “low concentration” in the result section (page 10), and the corresponding sentence now reads as: “At this concentration of ^{15}N -labelled tRNA, reasonable signal-to-noise ratio (SNR) is achieved with an acquisition time of 2 hours [...]”

Finally, we have added a statement about the total tRNAs concentration in the method section of the revised version page 26 with the above mentioned references.

Minor point: please straighten out the logic if the T54-m158 discussion at the bottom of page 11. The current explanation is very convoluted, and certain passages are unclear.

We have completely rephrased the T54-m¹A58 discussion at the bottom of page 11 to improve its clarity. The corresponding paragraph now reads as: *“However, our data suggest that the m¹A58 modification is strictly introduced after T54. Indeed, we detect the m¹A58 modification by its indirect effect on the T54 imino group signal (Fig. 3c). If the m¹A58 modification is introduced on a tRNA lacking the T54 modification, it would be detected by a large perturbation on the U54 imino group that would then resonate at a different position than that of T54 ($\delta_H = 12.27$ ppm; $\delta_N = 158.2$ ppm) since it lacks the T54 modification. Importantly, no unidentified transient signal, which could correspond to tRNAs with m¹A58 but still lacking T54, is observed during the incubations. The m¹A58 modification of U54-containing tRNAs is therefore not observed in our experiments, suggesting a strict introduction of m¹A58 after T54.”*

Reviewer #3

A growing theme in the RNA modification field is the idea that many modifications may not occur independent of each other but rather follow specific routes, whereby a pioneer modification may be important for a subsequent one. Implicit in such a cascade is the high potential for regulation. However, despite much progress in improving detection methods for various modifications, it is still technically challenging to map modifications to specific positions in a tRNA molecule and indeed mass spectrometry is the golden standard. In the present study, Barraud et al. implement the use of time-resolved NMR to monitor the temporal order of modifications in tRNA. Although the work is carefully performed and it has its merits, there are major issues with the work, which in my opinion limits the approach as applied here.

1. The use of NMR to analyze modifications is not necessarily new, for instance follow the work of Paul Agris and independently Darrel Davis' groups through the years. Here, the new thing is the use of time-resolved NMR, clearly the authors have successfully provided positional information about various modifications, however, the limitations are not from the detection itself, but from the idea that mixing an isotopically labeled tRNA, which I assume is folded somehow, into extracts will accurately say anything about the temporal nature of modifications. The very fact that the tRNA may or may not be folded argues against the reality of their temporal measurements and what can really be gleaned from those measurements. For instance, how do we know that the order established here is really a reflection of the modification pathway and not rather the struggle for a certain enzyme that may act early on the pathway to modify a “fully” folded transcript. Here, a good case can be made with m2G10, since methylations at the 9 position are important for tRNA folding, it stands to reason that such methylations may occur early in the modification/folding pathway and there is evidence that it occurs co-transcriptionally. Thus it is impossible to think that pseudouridine 55 will occur before G9 methylations. Perhaps highlighting the shortcomings of this approach.

There are several points here.

The first point concerns the potential limitations of our approach, in which the transcription process is dissociated from the modification process. As noted by reviewer #2, we are fully aware of the potential caveats of our methodology. This is why we have proposed here to validate our

conclusions by following a reverse genetic approach. We agree with this reviewer and with reviewer #2 that even though they do not present a detrimental effect to the performance of the technique, the potential shortcomings of our approach are worth to be mentioned and discussed.

As already mentioned, in order to follow this reviewer suggestion, as well as that of reviewer #2 (see above), we have expanded a sentence in the discussion page 20-21, which now reads as: *“even though our NMR approach might be biased by the lack of compartmentalisation, by the use of a relatively high concentration of substrate tRNA (40 μM) that stretches the modification process over an artificial time span, by an intrinsic decay of enzymatic activities during incubation that is likely enzyme-dependent, and by the fact that the transcription is here dissociated from the modification process, it enables the identification of connections between modifications.”*

The second point deals with the m²G10 modification, methylation at position 9 and the Ψ55 modification. We believe there is some misunderstanding here. In the yeast tRNA^{Phe}, the substrate used in our study, there is no modifications at position 9 (modifications at this position are introduced in yeast by Trm10 to yield m¹G9). There is, however, a modification at position 10, namely m²G10, which is introduced in yeast by the Trm11/Trm112 heterodimer. In this manuscript, we therefore do not provide any information and we do not discuss anything regarding the order of the introduction of modification at position 9 and 55.

2. Another major issue deals with a number of extrapolations made with the analysis and conclusions about the results that are either not granted by the study or are simply factually wrong. For example, several times the authors mention an enzyme being “less active” or “more active” without providing any evidence of the specific activity of the enzyme in question, or any measurement of enzymatic rates. Along these lines, on page 12 the statement is made that “it is difficult to discriminate between authentic dependence and a sequential order...” this is simply not correct. Careful analysis by mechanistic enzymology can easily tease out reaction order and the enzyme mechanism.

We thank the reviewer for pointing this aspect. Our formulation page 12 was not clear and was thus not understood in the way we wanted. We did not want to say that “it is difficult to discriminate between authentic dependence and a sequential order [...]” *in general*, but that it is difficult with our experimental setup. This is then one of the reasons why we have performed experiments with mutant strains.

We have rephrased our statement that now reads as: *“Importantly, with our experimental setup, it is difficult to discriminate between an authentic dependence on the prior introduction of certain modifications and a sequential order caused by different intrinsic catalytic activities of the enzymes.”*

In addition, we agree that, as stated by this reviewer, careful enzymology can be done with purified enzymes and tRNA substrates bearing or not pre-existing modifications. We consider that such approach, although very relevant, go beyond the scope of the present study. We clearly believe that it is something that could be conceived and performed if yeast enzymes can be produced and purified in their active form. In addition, we foresee that it could be combined with suggestions made by reviewer #1 and the use of different type of labelling, including the ¹³C labelling of the methyl donor SAM cofactor. We thank reviewer #3 for this constructive idea that could be implemented in a subsequent study on these aspects. We have added few sentences to discuss these aspects page 18-19: *“To discriminate direct from indirect effects is an important point that could be addressed in vitro by performing careful enzymology with purified enzymes and tRNA substrates bearing or not pre-existing modifications. Among other techniques, this*

approach could be implemented with NMR using different types of labelling, e.g. ¹⁵N-labelling of the tRNA substrate and a similar experimental setup as presented here, or ¹³C-labelling of the transferable methyl of the SAM cofactor and a monitoring of the modification process with ¹H-¹³C correlation spectra showing the incorporation of the methyl groups into the tRNA substrate.”

3. Further limitations of the system are provided by the fact that their model substrate, tRNAPhe has yW at position 37. The forcible precursor to yW is the methylated nucleotide m¹G37, which happens independent of the other wybutosine enzymes and yet no trace of m¹G37 is shown. It is not clear why this discrepancy.

Here the question is related to the fact that with the ¹⁵N-labelling adopted in this study, no information is available for the anticodon loop, as explained in the text page 9. We do not have any information on modifications at position 37, and for this reason, we do not report on m¹G37 as an intermediate along the wybutosine synthesis pathway. Study of modifications in the anticodon loop could potentially be achieved by using other types of labelling (namely ¹³C-labelling) as suggested by reviewer #1 (see answer to reviewer #1 point 2). We haven't so far investigated in detail these possibilities.

Reviewers' Comments:

Reviewer #1:

Remarks to the Author:

The authors addressed all my original concerns. The manuscript can now be published as is.

Reviewer #2:

Remarks to the Author:

The authors have supplied extensive responses to all issues raised by the reviewers. In particular, the arguments in their response to my principle criticism concerning time frame of kinetics and tRNA concentrations in cellular extracts, are convincing. I believe that this is an important piece of work that should be published with all due speed.

Reviewer #3:

Remarks to the Author:

The authors have perfectly addressed all my concerns. I recommend publication in its present form